# Improved Stochastic Optimization of LogSumExp

**Egor Gladin** [1]  **Alexey Kroshnin** [2]  **Jia-Jie Zhu** [3]  **Pavel Dvurechensky** [2]

## Abstract

The LogSumExp function, dual to the Kullback-Leibler (KL) divergence, plays a central role in many important optimization problems, including entropy-regularized optimal transport (OT) and distributionally robust optimization (DRO). In practice, when the number of exponential terms inside the logarithm is large or infinite, optimization becomes challenging since computing the gradient requires differentiating every term. We propose a novel convexity- and smoothness-preserving approximation to LogSumExp that can be efficiently optimized using stochastic gradient methods. This approximation is rooted in a sound modification of the KL divergence in the dual, resulting in a new $f$-divergence called the *Safe KL divergence*. Our experiments and theoretical analysis of the LogSumExp-based stochastic optimization, arising in DRO and continuous OT, demonstrate the advantages of our approach over existing baselines.

## 1. Introduction

Optimization problems arising in various fields involve the LogSumExp function, or, more generally, the log-partition functional

$$F(\varphi; \mu) := \log \int e^{\varphi(x)} \, \mathrm{d}\mu(x) \in (-\infty, \infty] \qquad (1)$$

mapping a measurable function $\varphi$ to $(-\infty, \infty]$ based on a probability measure $\mu$. The goal in such optimization problems is to minimize an objective involving $F$ w.r.t. $\varphi$ over some class.

LogSumExp function appears commonly in optimization objectives, e.g., multiclass classification with softmax prob-

---

[1] HSE University, Moscow, Russia [2] WIAS, Berlin, Germany [3] Department of Mathematics, KTH Royal Institute of Technology, Stockholm, Sweden. Correspondence to: Egor Gladin <elgladin@hse.ru>.

*Proceedings of the 43rd International Conference on Machine Learning*, Seoul, South Korea. PMLR 306, 2026. Copyright 2026 by the author(s).

abilities (Bishop, 2006), semi-dual formulation of entropy-regularized optimal transport (OT) (Peyré & Cuturi, 2019; Genevay et al., 2016), minimax problems (Pee & Royset, 2011), distributionally robust optimization (DRO) (Hu & Hong, 2012; Ben-Tal et al., 2013; Kuhn et al., 2025), maximum likelihood estimation (MLE) for exponential families and graphical models (Wainwright et al., 2008), variational Bayesian methods (Khan & Nielsen, 2018; Khan & Rue, 2023), information geometry (Amari & Nagaoka, 2000), KL-regularized Markov decision processes (Tiapkin et al., 2024). These problems involve minimizing $F(\varphi; \mu)$ w.r.t. a function $\varphi$, potentially parameterized by a vector $\theta$, e.g., a vector of neural network weights. Such optimization is characterized by two challenges. First, the decision variable $\varphi$ or $\theta$ often has large or infinite dimension. Second, the support of the measure $\mu$ can also be large or infinite. The first challenge is usually addressed by the use of first-order methods like Stochastic Gradient Descent (SGD), which are suitable for high-dimensional problems due to their cheap iterations. The second challenge is more delicate. Indeed, if $F(\varphi; \mu)$ is approximated by a large finite sum, then computing the exact gradient requires differentiating all exponential terms. Replacing this full aggregation by a sampled subset generally leads to biased gradient estimators (Lin et al., 2025).

In the current work, we propose a general-purpose approach to tackle both mentioned challenges. To that end, we use a SoftPlus approximation of $F(\varphi; \mu)$ that moves the expectation outside the logarithm, which allows using stochastic gradient methods while remaining close to the original objective. We start with a variational formulation analogous to the one in the Gibbs principle, but with the KL-divergence replaced with another $f$-divergence – the *(Overflow-)Safe KL* divergence. The corresponding variational problem can be of interest itself, as it possesses some properties which can be beneficial compared to the KL penalty – e.g., uniform density bound. Moreover, it can be also viewed as an approximation of a conditional value at risk functional (CVaR). In fact, the same functional (with different parameters) appeared in (Soma & Yoshida, 2020) in the context of smooth CVaR approximation. Thus, we demonstrate that it generates a family of problems including CVaR and LogSumExp minimization as limit cases.

**Related works.** The LogSumExp functional (1) has ap-

peared in many applications and has often been treated on a case-by-case basis. Bouchard (2007) study three upper bounds on LogSumExp for approximate Bayesian inference; one of them is a particular case of the approximation proposed in the present work. Titsias (2016) construct a bound on softmax probabilities and show that it leads to a bound on LogSumExp in the context of multiclass classification. Nielsen & Sun (2016) approximate LogSumExp in the context of estimating divergences between mixture models, combining bounds based on min and max. Tucker et al. (2017); Luo et al. (2020) propose and study unbiased estimators for latent-variable models based on Russian Roulette truncation. Lyne et al. (2015); Spring & Shrivastava (2017) focus on estimating the partition function itself, rather than on optimization problems involving the partition function.

KL-regularized and KL-constrained DRO provide important finite-sum instances of LogSumExp optimization. Hu & Hong (2012) and, subsequently, Levy et al. (2020) study DRO problems with $f$-divergences. When the ambiguity set is the unit simplex and a KL-divergence penalty is used, the resulting objective is the LogSumExp of the losses over the entire dataset. The batch-based approximation of Levy et al. (2020) replaces this objective with an average of LogSumExp terms computed on individual batches, which introduces a bias that can be reduced by significantly increasing the batch size.

Another closely related line of work treats KL-regularized DRO and other log-expectation-exponential objectives through stochastic compositional optimization. In particular, Qi et al. (2021) reformulate KL-regularized DRO as a compositional optimization problem and develop online stochastic methods, a perspective that was subsequently extended to KL-constrained DRO (Qi et al., 2023), broader finite-sum coupled compositional optimization problems (Wang & Yang, 2022), and LogSumExp-type objectives arising in contrastive learning (Yuan et al., 2022).

Our approach is complementary to this line of work. Rather than introducing a new optimizer for the original finite-sum LogSumExp objective, we modify the variational formulation itself by replacing the KL penalty with the Safe KL divergence. This yields a smooth approximation of the log-partition functional whose stochastic gradients are unbiased for the approximating objective and whose weights are uniformly bounded. Consequently, the basic convergence analysis avoids both the batch-size-dependent bias of sampled LogSumExp approximations and the exponential constants that may arise when optimizing objectives containing raw exponentials. The construction also applies naturally beyond finite sums, including continuous-measure settings where the log-partition functional is defined by an integral.

Deterministic LogSumExp *maximization* and minimization

were considered in Selvi et al. (2020) and Kan et al. (2024), respectively. For stabilizing numerical evaluation of the LogSumExp function, we refer to Blanchard et al. (2021); Higham (2021).

**Contributions.** Our main contributions are as follows:

1. We introduce a general-purpose and computationally efficient approach for handling the LogSumExp function in large-scale optimization problems by proposing a novel relaxation of this function. The proposed relaxation preserves key properties of the original LogSumExp function, such as convexity and smoothness, and turns the problem into an expectation minimization problem amenable to standard SGD-type methods. Furthermore, our method only requires a simple and tunable scalar parameter, allowing the relaxation to be made arbitrarily close to the original LogSumExp objective as desired.

2. We provide the theoretical backbone of this approximation, demonstrating that it is based on a modified version of the KL-divergence in the dual formulation. We term the resulting $f$-divergence the *Safe KL* divergence. It can be applied to various applications where KL-divergence is used.

3. We empirically evaluate the method on continuous entropy-regularized OT and several DRO formulations. The proposed method shows competitive performance in comparison to representative application-specific baselines and circumvents the overflow issue (Remark 3.1). It can also be combined with existing techniques. Therefore, it serves as a versatile tool for solving large-scale optimization problems.

4. Additionally, we provide insights into a few remarkable connections between the proposed approximation and existing notions such as the conditional value-at-risk.

**Notation.** Given $a, a_1, \ldots, a_n \in \mathbb{R}$, we define

$$\mathrm{LogSumExp}(a_1, \ldots, a_n) := \log\Big(\sum_{i=1}^{n} e^{a_i}\Big)$$

and $\mathrm{SoftPlus}(a) := \log(1 + e^a)$. Given a measurable space $\mathcal{X}$, by $\mathcal{P}(\mathcal{X})$ we denote the space of probability measures on $\mathcal{X}$, and by $\mathcal{C}(\mathcal{X})$ the space of continuous functions on $\mathcal{X}$. For $\mu, \nu \in \mathcal{P}(\mathcal{X})$ the Kullback–Leibler (KL) divergence is

$$D_{KL}(\mu, \nu) := \begin{cases} \int_{\mathcal{X}} \log \frac{\mathrm{d}\mu}{\mathrm{d}\nu}(x)\, \mathrm{d}\mu(x) & \mu \ll \nu, \\ +\infty & \text{otherwise,} \end{cases}$$

where $\log$ is the natural logarithm and $\mu \ll \nu$ denotes that $\mu$ is absolutely continuous w.r.t. $\nu$.

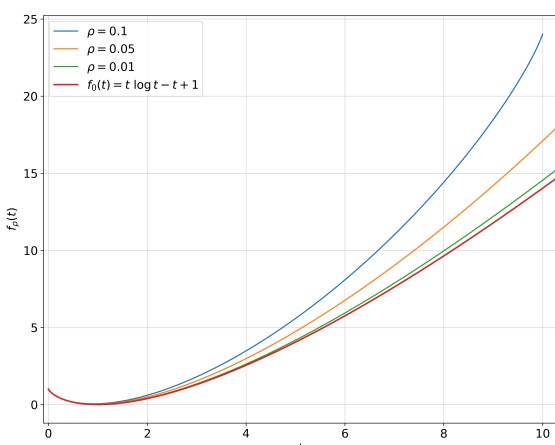

Figure 1. $f_\rho(t)$ for different values of $\rho$.

## 2. SoftPlus Approximation of Log-Partition Function

In this section, we present our approximation to the log-partition function (1) and describe its theoretical properties. Recall that by the Gibbs variational principle

$$F(\varphi; \mu) = \sup_\nu \left\{ \int_{\mathcal{X}} \varphi(x)\, d\nu(x) - D_{KL}(\nu, \mu) : \right.$$
$$\left. \nu \in \mathcal{P}(\mathcal{X}), \int_{\mathcal{X}} |\varphi(x)|\, d\nu(x) < \infty \right\} \quad (2)$$

with the maximum attained at the Gibbs measure $d\nu^*(x) = e^{\varphi(x) - F(\varphi;\mu)} d\mu(x)$, once $F(\varphi;\mu) < \infty$, see (Gibbs, 1902, Chapter XI, Theorem VI) or (Polyanskiy & Wu, 2025, Proposition 4.7) for the modern treatment.

We are going to construct an approximation of $F$ with better regularity properties by changing $D_{KL}$ to another $f$-divergence. Specifically, for any $0 < \rho < 1$, let us define the following.

**Definition 2.1** (Safe KL entropy). We define the Safe KL entropy generator $f_\rho \colon [0, \infty) \to \mathbb{R}$ by

$$f_\rho(t) := \begin{cases} t \log t + 1 + \frac{1 - \rho t}{\rho} \log(1 - \rho t), & 0 \le t \le \frac{1}{\rho}, \\ +\infty, & \text{otherwise.} \end{cases} \quad (3)$$

The resulting $f_\rho$-divergence, which we refer to as the *Safe KL divergence*, is given by

$$D_\rho(\nu, \mu) := \begin{cases} \int_{\mathcal{X}} f_\rho\left(\frac{d\nu}{d\mu}(x)\right)\, d\mu(x), & \nu \ll \mu, \\ +\infty, & \text{otherwise.} \end{cases} \quad (4)$$

Clearly, $f_\rho(t) \to f_0(t) := t \log t + 1 - t$ as $\rho \to 0$. Since $f_0$ induces the standard KL-divergence, $D_\rho$ is its approximation with accuracy regulated by the parameter $\rho$.

Using the variational representation, we define

$$F_\rho(\varphi; \mu) := \sup_\nu \left\{ \int_{\mathcal{X}} \varphi(x)\, d\nu(x) - D_\rho(\nu, \mu) : \right.$$
$$\left. \nu \in \mathcal{P}(\mathcal{X}), \int_{\mathcal{X}} |\varphi(x)|\, d\nu(x) < \infty \right\}. \quad (5)$$

(i.e., $F_\rho(\cdot; \mu)$ is the convex conjugate of $D_\rho(\cdot, \mu)$). Note that the last term in $f_\rho$ prevents the density $\frac{d\nu}{d\mu}$ from being too large. In particular, it cannot be greater than $\frac{1}{\rho}$. This can make the Safe KL divergence a reasonable choice for unbalanced OT or DRO, as it imposes a hard constraint on the reweighting unlike the standard $D_{KL}$. Moreover, it can also be used instead of the entropy penalization in regularized OT (cf. capacity constrained transport in (Benamou et al., 2015, section 5.2)).

Again, by the convex duality and the variational principle (see Birrell et al., 2022, Theorem 6), we state the following properties.

**Lemma 2.2.** *The functional $F_\rho$ defined by (5) has an equivalent variational representation*

$$F_\rho(\varphi; \mu) = \inf_{\alpha \in \mathbb{R}} \alpha + \int_{\mathcal{X}} f_\rho^*(\varphi(x) - \alpha)\, d\mu(x).$$

It is straightforward to check the following.

**Lemma 2.3.** *The conjugate function to $f_\rho$ is a rescaled SoftPlus, specifically,*

$$f_\rho^*(s) := \sup_{t \in \mathbb{R}_+} st - f_\rho(t) = \frac{1}{\rho} \log(1 + \rho e^s) - 1.$$

*Therefore, we obtain*

$$F_\rho(\varphi; \mu) = \inf_{\alpha \in \mathbb{R}} \alpha - 1 + \frac{1}{\rho} \int_{\mathcal{X}} \log\left(1 + \rho e^{\varphi(x) - \alpha}\right)\, d\mu(x). \quad (6)$$

In essence, we have replaced the exponential function with a rescaled SoftPlus. Furthermore, it is easy to see that the optimal $\alpha_\rho = \alpha_\rho(\varphi; \mu)$ satisfies

$$\int_{\mathcal{X}} \frac{e^{\varphi(x) - \alpha_\rho}}{1 + \rho e^{\varphi(x) - \alpha_\rho}}\, d\mu(x) = 1, \quad (7)$$

in particular, $\alpha_\rho < F(\varphi; \mu)$. Moreover, the maximum in (5) is attained at $d\nu_\rho^*(x) = \frac{e^{\varphi(x) - \alpha_\rho}}{1 + \rho e^{\varphi(x) - \alpha_\rho}}\, d\mu(x)$. Note that $0 < \frac{d\nu_\rho^*(x)}{d\mu(x)} < \frac{1}{\rho}$, which is due to the fact that the derivative of $t \log t$ explodes at $0$, preventing reaching the constraint.

The next proposition (proved in Appendix A) ensures that $F_\rho$ is a valid approximation of $F$.

**Proposition 2.4.** *Let $\mu \in \mathcal{P}(\mathcal{X})$ and $\varphi$ be a measurable function on $\mathcal{X}$.*

(i) *For all $0 < \rho \le \rho' < 1$, it holds $F_{\rho'}(\varphi; \mu) \le F_{\rho}(\varphi; \mu)$.*

(ii) *As $\rho \to 0+$, $F_{\rho}(\varphi; \mu) \to F_0(\varphi; \mu) := F(\varphi; \mu)$.*

(iii) *If $F(2\varphi; \mu) < \infty$, denote $\varkappa(\varphi; \mu) := e^{F(2\varphi) - 2F(\varphi)}$, then it holds for any $\rho \in \left(0, \frac{1}{\varkappa(\varphi; \mu)}\right)$*

$$F_{\rho}(\varphi; \mu) \ge F(\varphi; \mu) + \frac{\rho}{2} + \log(1 - \rho\varkappa(\varphi; \mu)). \quad (8)$$

(iv) *If $\varphi(x) \le M$ for all $x \in \mathcal{X}$, then*

$$F_{\rho}(\varphi; \mu) \ge F(\varphi; \mu) - \rho e^{M - F(\varphi; \mu)}$$

*for any $\rho \in \left(0, e^{F(\varphi; \mu) - M}\right)$.*

In particular, (i) and (iii) show that $F - O(\rho) \le F_{\rho} \le F$, and thus the parameter $\rho$ allows one to control the approximation accuracy. In the case of LogSumExp, the above proposition yields the following simple bounds.

**Corollary 2.5.** *Let $a_1, \ldots, a_n \in \mathbb{R}$. Then for any $\rho \in (0, 1)$*

$$\text{LogSumExp}(a_1, \ldots, a_n) - \rho \le$$

$$\le \inf_{\alpha \in \mathbb{R}} \alpha - 1 + \frac{1}{\rho} \sum_{i=1}^{n} \log(1 + \rho e^{a_i - \alpha})$$

$$\le \text{LogSumExp}(a_1, \ldots, a_n).$$

For $\rho = 1$ our approximation coincides with Bouchard's bound for LogSumExp (Bouchard, 2007).

### 2.1. Links to CVaR

Recall that the conditional value at risk (CVaR) w.r.t. a probability measure $\mu \in \mathcal{P}(\mathcal{X})$ at level $\rho \in (0, 1)$, associated with a function $\varphi$, can be defined (in the case of continuous distribution) as

$$\text{CVaR}_{\rho}(\varphi; \mu) := \mathbb{E}_{X \sim \mu}\left[\varphi(X) | \varphi(X) \ge Q_{1-\rho}\right]$$

$$= \frac{1}{\rho} \int_{\varphi(x) \ge Q_{1-\rho}} \varphi(x) \, d\mu(x),$$

where $Q_{1-\rho}$ is the $(1-\rho)$-quantile of $\varphi(X)$, $X \sim \mu$ (Rockafellar et al., 2000). Moreover, by Theorem 1 of Rockafellar et al. (2000), CVaR also has the following formulation:

$$\text{CVaR}_{\rho}(\varphi; \mu) = \inf_{\alpha \in \mathbb{R}} \alpha + \frac{1}{\rho} \int_{\mathcal{X}} (\varphi(x) - \alpha)_+ \, d\mu(x). \quad (9)$$

Remarkably, in (Soma & Yoshida, 2020) the authors obtained a smooth approximation to CVaR which, up to an additive constant, has the same form as $F_{\rho}$. However, they considered the approximation w.r.t. a different parameter—a "temperature" inside SoftPlus. Finally, Levy et al. (2020) proposed another similar smoothed version of CVaR (KL-regularized CVaR) in the context of DRO. For our approximation, we obtain the following bounds.

**Proposition 2.6.** *For all $0 < \rho < 1$ and $\lambda > 0$*

$$\text{CVaR}_{\rho}(\varphi; \mu) + \lambda(\log \rho - 1) \le \lambda F_{\rho}(\varphi/\lambda; \mu) \le$$

$$\le \text{CVaR}_{\rho}(\varphi; \mu) + \lambda\left(\log \rho - 1 + \frac{1}{\rho}\right). \quad (10)$$

### 2.2. The Case of Parametric Models

In some applications, the function $\varphi$ is defined as a parametric loss function $L(x, \theta)$, and the goal is to minimize an objective involving (1) w.r.t. $\theta$ to find the best model from the parametric family. For notational convenience, we write

$$F(\theta) := F(L(\cdot, \theta); \mu) = \log \int_{\mathcal{X}} e^{L(x, \theta)} \, d\mu(x),$$

and the problem of interest reads as

$$F^{\star} := \min_{\theta \in \Theta} F(\theta),$$

where $\Theta \subset \mathbb{R}^d$ is a nonempty compact convex parameter set. Combining our approximation (6) and the minimization w.r.t. parameter $\theta$, we obtain the following minimization problem

$$\min_{\theta \in \Theta, \alpha \in \mathcal{A}} G_{\rho}(\theta, \alpha), \quad (11)$$

where

$$G_{\rho}(\theta, \alpha) := \alpha - 1 + \frac{1}{\rho} \int_{\mathcal{X}} \log\left(1 + \rho e^{L(x, \theta) - \alpha}\right) \, d\mu(x),$$

and $\mathcal{A} \subseteq \mathbb{R}$ is an interval to be specified later. Clearly, $G_{\rho}$ is convex in $\alpha$. Moreover, if $L$ is convex in $\theta$ for $\mu$-a.e. $x$, then $G_{\rho}$ is jointly convex, meaning that our approximation preserves convexity.

Note that

$$f_{\rho}(t) = \frac{1}{\rho}\left((\rho t)\log(\rho t) + (1 - \rho t)\log(1 - \rho t)\right) +$$

$$+ 1 - t\log \rho.$$

Thus, unlike the KL entropy function $t\log t + 1 - t$, $f_{\rho}$ possesses the following favorable properties:

**Lemma 2.7.** *The entropy function $f_{\rho}$ is $\rho$-strongly convex. Its conjugate function $f_{\rho}^*$ is $\frac{1}{\rho}$-smooth.*

The above properties are useful from the computational optimization viewpoint. They imply, in particular, that the approximation preserves convexity and leads to a stochastic subgradient oracle with controlled second moment. Let $g(x, \theta) \in \partial_{\theta} L(x, \theta)$. Recall that

$$\frac{d}{dt}\log(1 + e^t) = \frac{e^t}{1 + e^t} =: \sigma(t),$$

hence for every $(\theta, \alpha)$, the vector

$$g_\rho(x, \theta, \alpha) := \begin{bmatrix} \rho^{-1}\sigma(L(x,\theta) - \alpha + \log\rho)\,g(x,\theta) \\ 1 - \rho^{-1}\sigma(L(x,\theta) - \alpha + \log\rho) \end{bmatrix}$$
(12)

is an unbiased stochastic subgradient of $G_\rho$, i.e.,

$$\mathbb{E}_{X \sim \mu}\big[g_\rho(X, \theta, \alpha)\big] \in \partial G_\rho(\theta, \alpha).$$

Moreover, since $0 \leq \sigma \leq 1$, for $0 < \rho \leq 1$,

$$\|g_\rho(x, \theta, \alpha)\|^2 \leq \rho^{-2}\|g(x,\theta)\|^2 + \rho^{-2}. \qquad (13)$$

Thus, a second-moment bound on the stochastic subgradients of $L$ implies a second-moment bound for the stochastic subgradients of $G_\rho$.

**Assumption 2.8.** $L(x, \cdot)$ is convex for every $x \in \mathcal{X}$; there exists a stochastic subgradient oracle $g(X, \theta) \in \partial_\theta L(X, \theta)$ satisfying $\mathbb{E}\|g(X,\theta)\|^2 \leq M^2$ for every $\theta \in \Theta$; numbers $\underline{F}$ and $U$ satisfy $\underline{F} \leq F^\star \leq U$; $\Theta^\star := \arg\min_{\theta \in \Theta} F(\theta)$ is nonempty, and

$$\hat{\varkappa} := \sup_{\theta \in \Theta} \varkappa(L(\cdot, \theta); \mu) < \infty.$$

**Theorem 2.9.** *Let Assumption 2.8 hold, and let* $\rho \in \left(0, \frac{1}{2\hat{\varkappa}}\right)$. *Suppose that a constant $D_\star$ satisfies* $\mathrm{dist}(\theta_1, \Theta^\star) \leq D_\star$. *Set*

$$\mathcal{A} := [\underline{F} - 2\hat{\varkappa}\rho, U],$$

*and define*

$$R^2 := D_\star^2 + (U - \underline{F} + 2\rho\hat{\varkappa})^2.$$

*After $N$ iterations of projected stochastic subgradient method for (11), initialized at* $(\theta_1, \alpha_1) \in \Theta \times \mathcal{A}$*, with stepsize*

$$\eta = \frac{\rho R}{\sqrt{N(M^2 + 1)}},$$

*the averaged iterate*

$$\begin{bmatrix} \bar{\theta}_N \\ \bar{\alpha}_N \end{bmatrix} := \frac{1}{N} \sum_{k=1}^{N} \begin{bmatrix} \theta_k \\ \alpha_k \end{bmatrix}$$

*satisfies*

$$\mathbb{E}[F(\bar{\theta}_N) - F^\star] \leq \frac{R\sqrt{M^2 + 1}}{\rho\sqrt{N}} + 2\rho\hat{\varkappa}.$$

*Remark* 2.10. When $L(x, \cdot)$ is Lipschitz-smooth and bounded from below, then $G_\rho$ is smooth on $\Theta \times (-\infty, a]$ for every $a \in \mathbb{R}$, see Proposition D.2.

## 3. Applications

In this section we consider several particular applications involving the objective (1) and show numerically, that our general-purpose approach based on approximation (6) leads to better performance of SGD-type algorithms than the baseline algorithms designed specifically for these applications. The source code for all experiments is available at https://github.com/egorgladin/logsumexp-approx.

### 3.1. Continuous Entropy-Regularized OT

The classical optimal transport (Monge–Kantorovich) problem consists in finding a coupling of two probability measures $\mu, \nu \in \mathcal{P}(\mathcal{X})$ which minimizes the integral of a given measurable cost function $c \colon \mathcal{X} \times \mathcal{X} \to \mathbb{R}_+$ (e.g., a distance), i.e.,

$$W(\mu, \nu) := \inf_{\pi \in \Pi(\mu,\nu)} \int c(x,y)\,\mathrm{d}\pi(x,y),$$

where $\Pi(\mu, \nu) \subset \mathcal{P}(\mathcal{X} \times \mathcal{X})$ is the set of couplings (transport plans) of $\mu$ and $\nu$ (see Kantorovich, 1942; Villani, 2008; Santambrogio, 2015). For simplicity of demonstration, we assume that the measures are defined on the same space $\mathcal{X}$, but the results extend trivially to the case of two different spaces. Following Cuturi (2013), we consider the entropy-regularized optimal transport (eOT) problem:

$$\min_{\pi \in \Pi(\mu,\nu)} \int_{\mathcal{X} \times \mathcal{X}} c(x,y)\,\mathrm{d}\pi(x,y) + \varepsilon D_{KL}(\pi, \mu \otimes \nu) \quad (14)$$

where $\mu \otimes \nu$ is the product measure. It is known that eOT admits the following dual and semi-dual formulations (see, e.g., (Genevay et al., 2016)):

$$W_\varepsilon(\mu, \nu) = \underbrace{\max_{u,v \in \mathcal{C}(\mathcal{X})} \iint_{\mathcal{X} \times \mathcal{X}} f_\varepsilon(x,y,u,v)\,\mathrm{d}\mu(x)\,\mathrm{d}\nu(y)}_{\text{dual}}$$

$$= \underbrace{\max_{v \in \mathcal{C}(\mathcal{X})} \int_{\mathcal{X}} h_\varepsilon(x,v)\,\mathrm{d}\mu(x)}_{\text{semi-dual}}, \qquad (15)$$

where

$$f_\varepsilon(x,y,u,v) := u(x) + v(y) - \\ - \varepsilon\exp\left(\frac{u(x) + v(y) - c(x,y)}{\varepsilon}\right), \quad (16)$$

$$h_\varepsilon(x,v) := \int_{\mathcal{X}} v(y)\mathrm{d}\nu(y) - \varepsilon - \\ - \varepsilon\log\left(\int_{\mathcal{X}} \exp\left(\frac{v(y) - c(x,y)}{\varepsilon}\right)\mathrm{d}\nu(y)\right), \quad (17)$$

and $\varepsilon > 0$ is the regularization coefficient. In the LSOT framework (Seguy et al., 2018), the potentials $u$ and $v$ are parameterized by neural networks and optimized via SGD.

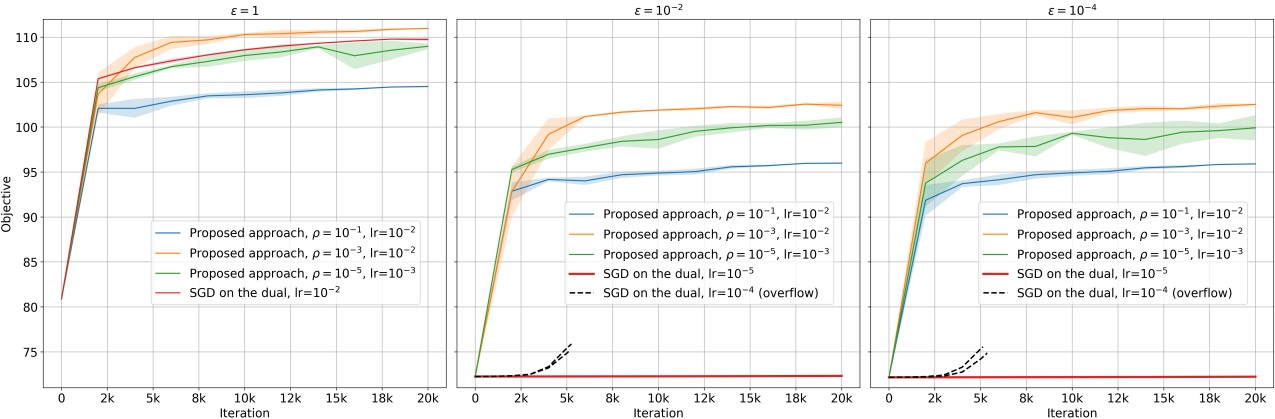

*Figure 2.* Test-set eOT semi-dual objective vs. iteration for different regularization strengths $\varepsilon$ (left to right: $1, 10^{-2}, 10^{-4}$). Lines show the mean across 5 runs; shaded areas are $\pm$ one standard deviation. We compare LSOT (red) with our method (colored by $\rho$). Dashed black curves are examples where LSOT with lr=$10^{-4}$ terminates early due to overflow, while lr=$10^{-5}$ results in a prohibitively slow convergence (nearly horizontal red lines for $\varepsilon = 10^{-2}, 10^{-4}$). Our proposed method remains stable and efficient for all $\varepsilon$.

While Appendix B.1 contains a more detailed literature review, we briefly position LSOT among other solvers to motivate its selection as a baseline. LSOT offers two key advantages relevant to our goals: it is **less computationally intensive** than modern solvers requiring adversarial training (Korotin et al., 2023; Gushchin et al., 2023; Asadulaev et al., 2024) or iterative Langevin dynamics (Mokrov et al., 2024), and it supports a **general cost function**—contrary to other efficient solvers like (Korotin et al., 2024) tailored to the quadratic cost. Therefore, to solve eOT with a general cost function under modest computational constraints, we adopt the LSOT framework as our primary baseline. In Appendix B.2 we also compare to Genevay et al. (2016), who use an RKHS parametrization for the potentials $u$ and $v$.

*Remark* 3.1 (The overflow issue). The main drawback of this approach is the presence of the exponent in the dual objective (and consequently in the SGD updates). Specifically, exponents are prone to floating-point exceptions (Goldberg, 1991), especially if the regularization parameter $\varepsilon$ is relatively small, which is often the case. For example, if $\varepsilon = 0.01$ and $z \geq 7.1$, then $e^{z/\varepsilon}$ exceeds the representable range of a *double-precision* (float64) floating-point number — an *overflow* happens. When single precision (float32) is used, an overflow happens even for $z \geq 0.89$.

**Our Approach.** If we consider instead the semi-dual formulation and use the approximation (6), we get the problem

$$\max_{v, \alpha \in \mathcal{C}(\mathcal{X})} \iint_{\mathcal{X} \times \mathcal{X}} \tilde{h}_\varepsilon(x, y, v, \alpha) \, \mathrm{d}\mu(x) \, \mathrm{d}\nu(y) \qquad (18)$$

with

$$\tilde{h}_\varepsilon(x, y, v, \alpha) := v(y) - \alpha(x) - $$
$$- \frac{\varepsilon}{\rho} \log\big(1 + \rho e^{(v(y) - c(x,y) - \alpha(x))/\varepsilon}\big) - \varepsilon, \quad (19)$$

which also admits neural network parameterization and optimization via SGD. One can show, in the same way as in Genevay et al. (2016), that this corresponds to the regularized OT problem (14) with the *Safe KL divergence $D_\rho$* rather than the usual KL, i.e.

$$\min_{\pi \in \Pi(\mu, \nu)} \int_{\mathcal{X} \times \mathcal{X}} c(x, y) \, \mathrm{d}\pi(x, y) + \varepsilon D_\rho(\pi, \mu \otimes \nu).$$

Note that this problem, in turn, can be viewed as a combination of the entropy-regularized and the capacity-constrained optimal transport. For $\rho > 0$, this approach is much more stable than the previous one when used in SGD. We illustrate this in the following experiments.

**Experiments.** We consider the MNIST (Deng, 2012) and EMNIST-letters (Cohen et al., 2017) datasets as samples from the distributions $\mu$ (digits) and $\nu$ (letters). Manhattan distance $\ell_1$ is chosen as the cost function for computing eOT between $\mu$ and $\nu$. We parameterize the functions $u, v$ in LSOT and $v, \alpha$ in our proposed approach using a multilayer perceptron with two hidden layers (dimensions 256 and 128) and ReLU activations. The batch size is 256, and the learning rate is selected via grid search over $\{10^{-6}, 10^{-5}, \dots, 10^{-1}\}$. The objective is evaluated on the empirical distributions of the dedicated test sets.

Figure 2 shows the performance of LSOT with the best learning rate for each regularization parameter $\varepsilon \in \{1, 10^{-2}, \dots, 10^{-4}\}$. It also depicts our proposed approach with the best learning rate for each $\rho \in \{10^{-1}, 10^{-3}, 10^{-5}\}$. The baseline performs adequately under strong regularization ($\varepsilon = 1$). However, for weaker regularization, a learning rate of $10^{-5}$ is required to avoid numerical instability, which leads to prohibitively slow progress (red curves). Increasing the rate to $10^{-4}$ (dashed black curves) results in numerical overflow after only $\approx$5k iterations, forcing us to abort the LSOT runs at that point.

The performance of our proposed approach aligns with the theoretical analysis in Section 2. A large $\rho$ yields stable convergence but introduces an approximation gap, while a very small $\rho$ degrades smoothness, necessitating a smaller step size and slower training. The intermediate value $\rho = 10^{-3}$ achieves the best trade-off, providing both accuracy and sufficient smoothness. In summary, our proposed approach to eOT is computationally efficient, accommodates general costs, and handles weak regularization robustly, thereby overcoming a key limitation of LSOT.

### 3.2. DRO with KL Divergence

One of the approaches to training a model that is robust to data distribution shifts and noisy observations is called Distributionally Robust Optimization (DRO) (Kuhn et al., 2025). In contrast to the standard Empirical Risk Minimization (ERM) approach, which minimizes the average loss on the training sample, DRO minimizes the risk for the worst-case distribution among those close to a reference measure (e.g., empirical distribution). A prominent example is KL divergence DRO (Hu & Hong, 2012), which is formulated as the saddle-point problem

$$\min_{\theta \in \Theta} \max_{p \in \Delta^n} \sum_{i=1}^n p_i \ell_i(\theta) - \lambda D_{KL}(p, \hat{p}), \qquad (20)$$

where $\theta \in \Theta$ is the model parameter vector, $\ell_i(\theta)$ is the respective loss on the $i$-th training example, $\Delta^n$ is the unit simplex in $\mathbb{R}^n$, $\hat{p} \in \Delta^n$ is the weight vector defining the empirical distribution (typically $\hat{p} = \frac{1}{n}\mathbf{1}$), and $D_{KL}$ is the Kullback–Leibler divergence which discourages distributions that are too far from the empirical one, $\lambda > 0$ is the penalty coefficient. For fixed $\theta$, the solution of the maximization problem is given by $p_i^*(\theta) \coloneqq \frac{e^{\ell_i(\theta)/\lambda}}{\sum_j e^{\ell_j(\theta)/\lambda}}$, which reduces the problem to

$$\min_{\theta \in \Theta} \mathcal{L}(\theta) \coloneqq \lambda \log\left(\frac{1}{n} \sum_{i=1}^n e^{\ell_i(\theta)/\lambda}\right). \qquad (21)$$

However, when $n$ is large, computing the full gradient $\nabla \mathcal{L}(\theta) = \sum_{i=1}^n p_i^*(\theta)\nabla \ell_i(\theta)$ becomes costly. A straightforward approach (Levy et al., 2020) is to sample a batch $D$, compute the respective softmax weights

$$p_i^D(\theta) \coloneqq \frac{e^{\ell_i(\theta)/\lambda}}{\sum_{j \in D} e^{\ell_j(\theta)/\lambda}},$$

and define a gradient estimator by

$$\tilde{\nabla}_D \mathcal{L}(\theta) = \sum_{i \in D} p_i^D(\theta)\nabla \ell_i(\theta). \qquad (22)$$

However, this introduces a bias and requires using large batch sizes to keep it sufficiently small.

**Our Approach.** Instead, we propose to use the approximation (6), which results in the problem

$$\min_{\substack{\theta \in \Theta \\ \alpha \in \mathbb{R}}} G(\theta, \alpha) \coloneqq \frac{1}{n} \sum_{i=1}^n \left\{ \alpha + \frac{\lambda}{\rho} \log\left(1 + \rho e^{(\ell_i(\theta) - \alpha)/\lambda}\right) \right\}. \qquad (23)$$

Like in the previous subsection, this can be interpreted as switching from $D_{KL}$ penalty in (20) to *Safe KL* $D_\rho$. The respective gradient estimators are

$$\tilde{\nabla}_\theta^D G(\theta, \alpha) \coloneqq \frac{1}{|D|} \sum_{i \in D} \sigma_\rho\left(\frac{\ell_i(\theta) - \alpha}{\lambda}\right) \nabla \ell_i(\theta),$$

$$\tilde{\nabla}_\alpha^D G(\theta, \alpha) \coloneqq 1 - \frac{1}{|D|} \sum_{i \in D} \sigma_\rho\left(\frac{\ell_i(\theta) - \alpha}{\lambda}\right). \qquad (24)$$

**Experiments.** Consider the California housing dataset (Pace & Barry, 1997) consisting of 20,640 objects represented by 8 features. Let $\ell_i$ be the squared error of a linear model, $\ell_i(\theta) = (y_i - \theta^\top x_i)^2$. We use accelerated SGD with the gradient estimator (22) (Levy et al., 2020) as the baseline approach for solving (21), and compare it to our proposed gradient estimator (24). We consider various values of the penalty coefficient $\lambda \in \{1/5, 1, 5\}$ and batch sizes $|D| \in \{10, 10^2, 10^3\}$. For each configuration, we select the optimal learning rate from $\{10^{-9}, 10^{-8}, \ldots, 10^{-4}\}$. The approximation accuracy parameter $\rho$ in our method is varied across $\{10^{-1}, 10^{-3}, 10^{-5}\}$. Momentum is fixed at 0.9 (without tuning), and the least squares solution is used as the initial point for optimization.

Numerical results are presented in Table 1, showing the objective value (mean $\pm$ standard deviation across 10 runs) after 50 epochs, where the methods typically reach a plateau. In each column, the best-performing configurations are highlighted in bold. For $\lambda = 1/5, |D| \in \{10, 10^2\}$, no results are displayed in bold as all configurations perform similarly. As seen from the table, the baseline and our estimator achieve comparable performance for large batch sizes ($|D| = 10^3$). However, for smaller batches, our method typically outperforms the baseline. Both approaches handle various $\lambda$ values well, with the exception of the baseline method combined with small batch sizes.

Regarding the approximation parameter $\rho$, large values ($\rho = 10^{-1}$) generally result in a noticeable approximation gap, while excessively small values ($\rho = 10^{-5}$) deteriorate the smoothness of the objective and consequently slow convergence. The intermediate value $\rho = 10^{-3}$ thus provides the best trade-off in this experiment, offering both good approximation accuracy and favorable optimization properties.

Additional KL-DRO results are reported in Appendix C.1. There, we evaluate the methods on an income-prediction task based on ACS PUMS data under an imbalanced train–

*Table 1.* Objective value (21) (mean $\pm$ std across 10 runs) at epoch 50 for baseline (22) (Levy et al., 2020) and proposed gradient estimator (24) with different $\rho$ values. Results are shown for various penalty coefficients $\lambda$ and batch sizes $|D|$, with optimal learning rates selected from $\{10^{-9}, \ldots, 10^{-4}\}$. Best results per column are shown in bold.

| Approach | $\lambda = 1/5$ | | | $\lambda = 1$ | | | $\lambda = 5$ | | |
|---|---|---|---|---|---|---|---|---|---|
| | $|D|{=}10$ | $|D|{=}10^2$ | $|D|{=}10^3$ | $|D|{=}10$ | $|D|{=}10^2$ | $|D|{=}10^3$ | $|D|{=}10$ | $|D|{=}10^2$ | $|D|{=}10^3$ |
| Baseline (22) | 26.9±0.7 | 15.6±6.0 | **9.1±4.6** | 20.0±0.9 | 5.2±2.9 | **2.3±0.2** | 0.87±0.01 | 0.88±0.00 | 0.79±0.00 |
| (24), $\rho{=}10^{-1}$ | 27.7±0.6 | 27.7±0.7 | 40.1±0.5 | 21.1±1.1 | 21.3±1.1 | 21.8±2.1 | 0.87±0.01 | 0.87±0.01 | 0.88±0.02 |
| (24), $\rho{=}10^{-3}$ | 21.2±9.8 | 18.6±7.7 | 25.3±0.1 | **2.1±0.0** | **2.1±0.0** | 2.5±1.2 | **0.76±0.02** | **0.78±0.00** | **0.78±0.00** |
| (24), $\rho{=}10^{-5}$ | 19.2±9.6 | 17.5±6.6 | 24.3±0.3 | 3.0±0.0 | 3.0±0.0 | 3.0±0.0 | 1.03±0.00 | 1.03±0.00 | 1.03±0.00 |

test split across states, which creates a distribution shift between training and testing populations. The results show that the lower DRO objective values achieved by the proposed estimator are also reflected in competitive or improved regression metrics, including worst-group RMSE and MAE.

### 3.3. DRO with Unbalanced OT

In the KL divergence DRO described in the previous subsection, uncertainty set is limited to distributions with the same support as the empirical measure $\mu = \frac{1}{n}\sum_i \delta_{x_i}$. Another popular approach, Wasserstein DRO (WDRO) (Mohajerin Esfahani & Kuhn, 2018; Sinha et al., 2018), considers the worst-case risk over shifts within a Wasserstein (OT) ball around a reference measure $\mu$ instead of the KL-ball in (20), thus including continuous probability measures. Unfortunately, this approach is not resilient to outliers that are geometrically far from the clean distribution since OT metric is sensitive to them (Nietert et al., 2023). A natural generalization is to switch to semi-balanced OT (Liero et al., 2018; Chizat et al., 2018; Kondratyev et al., 2016), which replaces a hard constraint on one of the marginals with a mismatch penalty function, e.g.,

$$W_\beta(\nu, \mu) = \inf_{\substack{\pi \in \mathcal{P}(\mathcal{X} \times \mathcal{X}) \\ \pi_1 = \nu}} \int_{\mathcal{X} \times \mathcal{X}} c(x, z)\, \mathrm{d}\pi(x, z) + \beta\, D_{KL}(\pi_2, \mu),$$

where $\pi_1$ and $\pi_2$ are first and second marginals of $\pi$, respectively, $\beta > 0$ is the marginal penalty parameter. Intuitively, this discrepancy measure allows to ignore some points (e.g., outliers) by paying a small price for mismatch in marginals. The (penalty-form) DRO problem can be written as

$$\min_{\theta \in \Theta} \max_{\nu \in \mathcal{P}(\mathcal{X})} \int_{\mathcal{X}} \ell(\theta, x)\, \mathrm{d}\nu(x) - \lambda W_\beta(\nu, \mu),$$

where $\lambda > 0$ is the Lagrangian penalty parameter. By standard duality, Wang et al. (2024) showed that when $\mu = \frac{1}{n}\sum_i \delta_{x_i}$ is the empirical distribution, this is equivalent to

$$\min_{\theta \in \Theta} F(\theta) := \gamma \log\left(\frac{1}{n}\sum_{i=1}^{n} e^{\hat{\ell}_i(\theta)/\gamma}\right)$$

$$\text{with } \hat{\ell}_i(\theta) := \sup_{z \in \mathcal{X}} \{\ell(\theta; z) - \lambda c(z, x_i)\}, \quad (25)$$

where $\gamma := \lambda\beta$. To avoid the costly gradient computation of LogSumExp, the authors drop the logarithm and use SGD to optimize the sum of exponents,

$$\min_{\theta \in \Theta} \frac{1}{n}\sum_{i=1}^{n} e^{\hat{\ell}_i(\theta)/\gamma}. \quad (26)$$

The major downside of this approach is that the exponent terms have a large variance, and SGD is prone to floating-point exceptions (overflow) unless a very small stepsize is tuned, which slows down the convergence and can be time-consuming and unstable in practice.

**Our Approach.** To overcome this issue, we propose leveraging the approximation (6), which leads to the problem

$$\min_{\substack{\theta \in \Theta \\ \alpha \in \mathbb{R}}} \frac{1}{n}\sum_{i=1}^{n} \left\{\alpha + \frac{\gamma}{\rho}\log\left(1 + \rho e^{(\hat{\ell}_i(\theta) - \alpha)/\gamma}\right)\right\}, \quad (27)$$

where $\rho > 0$ is a parameter controlling the accuracy of the approximation. This approximation can be efficiently optimized with SGD. Note that our method can also be applied to other DRO algorithms such as Sinkhorn DRO (Wang et al., 2025), which we omit to avoid redundancy.

**Experiments.** We consider MNIST dataset (Deng, 2012) with labels corrupted by feature-dependent noise (see Algan & Ulusoy, 2020) (noise ratio 25%). Let $\theta$ denote weights of a CNN with two convolutional layers (32 and 64 channels, kernel size 3, ReLU activations, and $2{\times}2$ max pooling), followed by a fully connected classifier with one hidden layer of 128 units, and let $\ell(\theta; z)$ be its cross entropy loss on object $z$. In the experiment, SGD with batch size 32 is applied to problems (26) (baseline) and (27) (proposed approach). We consider values of the stepsize $\eta \in \{10^{-9}, 10^{-8}, \ldots, 10\}$. For the inner maximization problem in (25), just 5 iterations of Nesterov's accelerated gradient method were sufficient to reach plateau in terms of the objective value.

Table 2 presents the final objective values (mean $\pm$ standard deviation across 5 runs) after 20 epochs, at which point the methods typically stop making significant progress. The best-performing configuration in each column is highlighted

*Table 2.* Objective value (25) (mean ± std across 5 runs) at epoch 20 for the baseline (Wang et al., 2024) and the proposed approach (with different $\rho$ values), i.e., SGD on (26) and (27), respectively. Results are shown for various penalty coefficients $\gamma$ and $\lambda$, with optimal learning rates selected from $\{10^{-9}, 10^{-8}, \ldots, 10\}$. Best results per column are shown in bold.

| Approach | $\gamma = 1/5$ | | | $\gamma = 1$ | | | $\gamma = 5$ | | |
|---|---|---|---|---|---|---|---|---|---|
| | $\lambda = 1/5$ | $\lambda = 1$ | $\lambda = 5$ | $\lambda = 1/5$ | $\lambda = 1$ | $\lambda = 5$ | $\lambda = 1/5$ | $\lambda = 1$ | $\lambda = 5$ |
| Baseline (26) | $1.79\pm0.01$ | $1.78\pm0.01$ | $1.77\pm0.01$ | $0.86\pm0.01$ | $0.81\pm0.01$ | $0.79\pm0.01$ | $0.33\pm0.06$ | $0.22\pm0.08$ | $0.17\pm0.02$ |
| (27), $\rho = 1$ | $4.76\pm0.68$ | $5.23\pm0.52$ | $6.04\pm0.79$ | $0.50\pm0.14$ | $0.42\pm0.40$ | $1.53\pm2.23$ | $0.23\pm0.03$ | $0.18\pm0.10$ | $\mathbf{0.09\pm0.03}$ |
| (27), $\rho = 10^{-1}$ | $1.57\pm0.16$ | $1.29\pm0.30$ | $1.93\pm0.92$ | $\mathbf{0.45\pm0.03}$ | $\mathbf{0.37\pm0.07}$ | $\mathbf{0.31\pm0.04}$ | $\mathbf{0.21\pm0.02}$ | $0.16\pm0.09$ | $0.12\pm0.06$ |
| (27), $\rho = 10^{-2}$ | $\mathbf{1.35\pm0.04}$ | $\mathbf{1.28\pm0.05}$ | $\mathbf{1.25\pm0.05}$ | $0.50\pm0.05$ | $0.50\pm0.11$ | $0.48\pm0.02$ | $\mathbf{0.21\pm0.03}$ | $\mathbf{0.12\pm0.03}$ | $0.26\pm0.04$ |

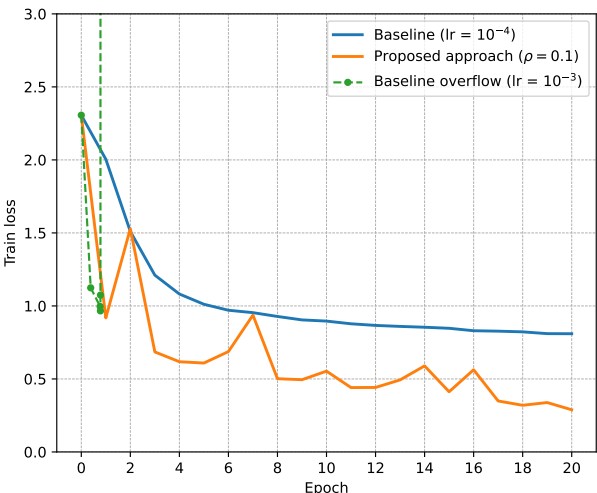

*Figure 3.* Example trajectories of the baseline with lr $= 10^{-4}$ (blue) and the proposed approach with $\rho = 0.1$, lr $= 10^{-2}$ (orange). Y-axis corresponds to the objective (25) with $\lambda = \gamma = 1$. Green dashed curve illustrates that the baseline diverges during the first epoch even for a relatively small lr $= 10^{-3}$.

in bold. The table shows that the proposed approach consistently outperforms the baseline. For the approximation parameter $\rho$, a large value ($\rho = 1$) often introduces a noticeable approximation gap. An exception occurs when the initial objective is sufficiently smooth, as with $\gamma = 5$. The value $\rho = 10^{-1}$ offers robust performance across most scenarios, whereas $\rho = 10^{-2}$ is better suited for ill-conditioned objectives (e.g., $\gamma = 1/5$) that demand high approximation accuracy.

To illustrate the optimization behavior underlying the aggregated results in Table 2, Figure 3 shows example trajectories for a fixed seed and parameter set ($\lambda = \gamma = 1$). The proposed approach with $\rho = 0.1$ (orange) minimizes the objective using a relatively large learning rate of $10^{-2}$. The baseline (blue), however, requires a learning rate of $10^{-4}$ to avoid numerical overflow, as demonstrated by its rapid divergence (green dashed curve) for lr $= 10^{-3}$. Consequently, the proposed method achieves lower objective values, while the baseline's small learning rate results in slow progress.

Additional UOT-DRO results are reported in Appendix C.3. In that experiment, we increase the label-noise ratio to 85%, which provides a more challenging setting and highlights that the improved objective values obtained by the proposed method can translate into better prediction metrics under severe label corruption.

## 4. Conclusion

We introduce a novel approximation to the log partition function, and in particular to LogSumExp, which arises in numerous applications across machine learning and optimization. In the dual formulation, this approximation corresponds to the Safe KL divergence. The proposed LogSumExp approximation preserves convexity and smoothness, admits unbiased stochastic gradients for the surrogate objective, and provides a controllable approximation bias independent of the batch size. Our empirical results highlight its practical advantages across tasks in continuous entropy-regularized OT and DRO, especially in regimes where standard exponential-based formulations suffer from numerical instability.

The method also has limitations. First, since Safe KL modifies the original KL divergence, application-specific structural properties of KL-based formulations may require separate analysis. Second, the parameter $\rho$ controls the bias–stability tradeoff and currently requires tuning; an adaptive procedure for selecting $\rho$ could further improve practicality. Finally, our experiments cover only a limited set of downstream tasks. Future work includes a deeper study of Safe KL formulations in specific applications and extensions to other settings where LogSumExp and KL duality play a central role.

## Impact Statement

This paper presents work whose goal is to advance the field of Machine Learning. Our main goal is to make ML applications where LogSumExp minimization is needed less computationally demanding. There are many potential societal consequences of our work, yet we don't see any direct consequences that we feel must be specifically highlighted

here.

## Acknowledgements

The work of Egor Gladin was supported by the grant for research centers in the field of AI provided by the Ministry of Economic Development of the Russian Federation in accordance with the agreement 000000C313925P4E0002 and the agreement with HSE University №139-15-2025-009

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

## A. Proofs for Section 2

*Proof of Proposition 2.4.* (i),(ii) Consider the function $g(t) := \frac{\log(1+t)}{t}$. It is decreasing and convex on $(0, \infty)$, $g(t) \to 1$ and $g'(t) \to -\frac{1}{2}$ as $t \to 0+$. Note that

$$F_\rho(\varphi; \mu) = \inf_{\alpha \in \mathbb{R}} \alpha - 1 + \int_{\mathcal{X}} e^{\varphi(x)-\alpha} g\left(\rho e^{\varphi(x)-\alpha}\right) \, d\mu(x).$$

Then (i) follows immediately from (6) and the monotonicity of $g$. The monotone convergence theorem yields (ii) since

$$F(\varphi; \mu) = \inf_{\alpha \in \mathbb{R}} \alpha - 1 + \int_{\mathcal{X}} e^{\varphi(x)-\alpha} \, d\mu(x).$$

Now, let us prove (iii). Consider the optimal $\alpha_\rho = \alpha_\rho(\varphi; \mu)$ satisfying (7). By Jensen's inequality

$$\int_{\mathcal{X}} \log\left(1 + \rho e^{\varphi(x)-\alpha_\rho}\right) \, d\mu(x) = -\int_{\mathcal{X}} \log\left(1 - \frac{\rho e^{\varphi(x)-\alpha_\rho}}{1 + \rho e^{\varphi(x)-\alpha_\rho}}\right) \, d\mu(x)$$

$$\geq -\log\left(1 - \int_{\mathcal{X}} \frac{\rho e^{\varphi(x)-\alpha_\rho}}{1 + \rho e^{\varphi(x)-\alpha_\rho}} \, d\mu(x)\right) = -\log(1 - \rho),$$

thus

$$F_\rho(\varphi; \mu) = \alpha_\rho - 1 + \frac{1}{\rho} \int_{\mathcal{X}} \log\left(1 + \rho e^{\varphi(x)-\alpha_\rho}\right) \, d\mu(x) \geq \alpha_\rho - 1 - \frac{\log(1 - \rho)}{\rho} \geq \alpha_\rho + \frac{\rho}{2}. \qquad (28)$$

It remains to get a lower bound on $\alpha_\rho$. By the monotonicity of $\frac{t}{1+t}$ we deduce that $\alpha_\rho \geq \alpha$ for any $\alpha$ such that

$$\int_{\mathcal{X}} \frac{\rho e^{\varphi(x)-\alpha}}{1 + \rho e^{\varphi(x)-\alpha}} \, d\mu(x) \geq \rho. \qquad (29)$$

Recall that if $X, Y$ are random variables and $Y > 0$, then $\mathbb{E}(X^2/Y) \geq (\mathbb{E}X)^2/(\mathbb{E}Y)$. Therefore,

$$\int_{\mathcal{X}} \frac{e^{\varphi(x)-\alpha}}{1 + \rho e^{\varphi(x)-\alpha}} \, d\mu(x) = \int_{\mathcal{X}} \frac{e^{2\varphi(x)-2\alpha}}{e^{\varphi(x)-\alpha} + \rho e^{2\varphi(x)-2\alpha}} \, d\mu(x) \geq \frac{(\int e^{\varphi(x)-\alpha} \, d\mu)^2}{\int (e^{\varphi(x)-\alpha} + \rho e^{2\varphi(x)-2\alpha}) \, d\mu} = \frac{u^2}{u + \rho e^{F(2\varphi)-2\alpha}},$$

where $u := \int e^{\varphi(x)-\alpha} \, d\mu = e^{F(\varphi)-\alpha}$. Note that

$$e^{F(2\varphi)-2\alpha} = e^{F(2\varphi)-2F(\varphi)} e^{2F(\varphi)-2\alpha} = u^2 e^{F(2\varphi)-2F(\varphi)}.$$

Denote $a := \rho e^{F(2\varphi)-2F(\varphi)}$, then

$$\frac{u^2}{u + \rho e^{F(2\varphi)-2\alpha}} = \frac{u^2}{u + u^2 a} = \frac{u}{1 + ua}.$$

If $a < 1$, i.e., $\rho < e^{2F(\varphi)-F(2\varphi)}$, take $\alpha$ such that $u = \frac{1}{1-a}$, then

$$\frac{u}{1 + ua} = \frac{1}{(1-a)(1 + \frac{a}{1-a})} = 1 \implies \alpha \text{ fulfills (29)} \implies \alpha_\rho \geq \alpha.$$

Also note that $u = \frac{1}{1-a} \iff -\log(1 - a) = F(\varphi) - \alpha$, so we get

$$\alpha_\rho \geq F(\varphi) + \log(1 - a) \geq F(\varphi) + \log(1 - \rho e^{F(2\varphi)-2F(\varphi)}). \qquad (30)$$

Combining this with (28) and using $F_\rho(\varphi; \mu) \leq F(\varphi; \mu)$, we arrive at (8).

(iv) Finally, let $\varphi(x) \leq M$ for all $x \in \mathcal{X}$. Then by concavity

$$\int_{\mathcal{X}} \log\left(1 + \rho e^{\varphi(x)-\alpha}\right) \, d\mu(x) \geq \int_{\mathcal{X}} e^{\varphi(x)-M} \log\left(1 + \rho e^{M-\alpha}\right) \, d\mu(x) = e^{F(\varphi;\mu)-M} \log\left(1 + \rho e^{M-\alpha}\right)$$

for all $\alpha \in \mathbb{R}$. Therefore,

$$
\begin{aligned}
F_\rho(\varphi; \mu) &\geq \min_\alpha \alpha - 1 + \frac{e^{F(\varphi;\mu)-M}}{\rho} \log\left(1 + \rho e^{M-\alpha}\right) \\
&= F(\varphi; \mu) - 1 - \frac{1 - \rho e^{M-F(\varphi;\mu)}}{\rho e^{M-F(\varphi;\mu)}} \log\left(1 - \rho e^{M-F(\varphi;\mu)}\right) \\
&\geq F(\varphi; \mu) - \rho e^{M-F(\varphi;\mu)}.
\end{aligned}
$$

Here we used the inequality

$$
\frac{1-t}{t} \log(1-t) \leq t - 1, \quad 0 < t < 1.
$$

$\square$

*Proof of Corollary 2.5.* Set $\mu_n \coloneqq \frac{1}{n} \sum_{i=1}^n \delta_{a_i} \in \mathcal{P}(\mathbb{R})$. Then

$$
\text{LogSumExp}(a_1, \ldots, a_n) = \log n + \log\left(\int_{\mathbb{R}} e^x \, \mathrm{d}\mu_n(x)\right) = \log n + F(id; \mu_n)
$$

and

$$
\begin{aligned}
\inf_{\alpha\in\mathbb{R}} \alpha - 1 + \frac{1}{\rho} \sum_{i=1}^n \log(1 + \rho e^{a_i-\alpha}) &= \inf_{\alpha\in\mathbb{R}} \alpha - 1 + \frac{n}{\rho} \int_{\mathbb{R}} \log(1 + \rho e^{x-\alpha}) \, \mathrm{d}\mu_n(x) \\
&= \inf_{\alpha\in\mathbb{R}} \alpha - 1 + \frac{n}{\rho} \int_{\mathbb{R}} \log\left(1 + \frac{\rho}{n} e^{x-\alpha+\log n}\right) \, \mathrm{d}\mu_n(x) \\
&= \log n + F_{\rho/n}(id; \mu_n).
\end{aligned}
$$

Since

$$
e^{F(id;\mu_n)-\max_i a_i} = \frac{\sum_{i=1}^n e^{a_i}}{n \max_i e^{a_i}} \geq \frac{1}{n} > \frac{\rho}{n},
$$

Proposition 2.4(i) and (iv) yields

$$
F(id; \mu_n) - \rho \leq F_{\rho/n}(id; \mu_n) \leq F(id; \mu_n).
$$

The claim follows. $\square$

*Proof of Proposition 2.6.* As $\lambda \log(1 + e^{t/\lambda}) > t_+ := \max\{0, t\}$, we get

$$
\begin{aligned}
\lambda F_\rho(\varphi/\lambda; \mu) &\geq \lambda \inf_{\alpha\in\mathbb{R}} \alpha - 1 + \frac{1}{\rho} \int_{\mathcal{X}} \left(\log\rho + \frac{\varphi(x)}{\lambda} - \alpha\right)_+ \, \mathrm{d}\mu(x) \\
&= \lambda(\log\rho - 1) + \inf_{\alpha\in\mathbb{R}} \alpha + \frac{1}{\rho} \int_{\mathcal{X}} (\varphi(x) - \alpha)_+ \, \mathrm{d}\mu(x).
\end{aligned}
$$

The infimum in the r.h.s. is the variational formula for CVaR (9), thus we get the first inequality in (10). The second inequality can be obtained in a similar way using that $\lambda \log(1 + e^{t/\lambda}) < t_+ + \lambda$. $\square$

*Proof of Lemma 2.7.* Recall that we have

$$
f_\rho(t) = \frac{1}{\rho}((\rho t)\log(\rho t) + (1 - \rho t)\log(1 - \rho t)) + 1 - t\log\rho.
$$

Simplifying, we obtain

$$
f_\rho(t) = t\log t + \frac{1}{\rho}(1 - \rho t)\log(1 - \rho t) + 1.
$$

The first derivative is calculated as follows:

$$
\frac{d}{dt}(t\log t) = \log t + 1, \qquad \frac{d}{dt}\left(\frac{1}{\rho}(1 - \rho t)\log(1 - \rho t)\right) = -\left(\log(1 - \rho t) + 1\right),
$$

so

$$f'_\rho(t) = (\log t + 1) - \big(\log(1 - \rho t) + 1\big) = \log t - \log(1 - \rho t) = \log\left(\frac{t}{1 - \rho t}\right).$$

The second derivative is calculated as follows:

$$\frac{d}{dt}(\log t) = \frac{1}{t}, \qquad \frac{d}{dt}(\log(1 - \rho t)) = -\frac{\rho}{1 - \rho t},$$

thus

$$f''_\rho(t) = \frac{1}{t} + \frac{\rho}{1 - \rho t} = \frac{1}{t(1 - \rho t)}.$$

By symmetry, we can see that the minimum value of the second derivative is achieved at $t^* = \frac{1}{2\rho}$, and it is equal to $4\rho$. Thus, for all $t \in \mathrm{dom} f_\rho$, we have that $f''_\rho(t) \geq 4\rho > \rho$. Thus, by (Nesterov, 2018, Theorem 2.1.11), $f_\rho$ is $\rho$-strongly convex. By (Zhou, 2018, Theorem 1) this also implies that its conjugate function $f^*_\rho$ is $\frac{1}{\rho}$-smooth. $\square$

*Proof of Theorem 2.9.* Projected stochastic subgradient method with stepsize $\eta$ satisfies (Nemirovski et al., 2009) for any feasible point $(\hat\theta, \hat\alpha) \in \Theta \times \mathcal{A}$

$$\mathbb{E}[G_\rho(\bar\theta_N, \bar\alpha_N) - G_\rho(\hat\theta, \hat\alpha)] \leq \frac{\|(\theta_1, \alpha_1) - (\hat\theta, \hat\alpha)\|^2}{2\eta N} + \frac{M_\rho^2 \eta}{2},$$

where $M_\rho^2 := \frac{M^2 + 1}{\rho^2}$ is a uniform second-moment bound on the stochastic subgradient of $G_\rho$ due to (13). Since $\mathrm{dist}(\theta_1, \Theta^\star) \leq D_\star$, there exists $\theta^\star \in \Theta^\star$ such that

$$\|\theta_1 - \theta^\star\| \leq D_\star.$$

Let us take

$$(\hat\theta, \hat\alpha) := (\theta^\star, \alpha_\rho(\theta^\star)), \qquad \alpha_\rho(\theta^\star) \in \arg\min_{\alpha \in \mathbb{R}} G_\rho(\theta^\star, \alpha).$$

Using the fact that

$$\log(1 - a) \geq -2a \quad \text{for any } a \in (0, 1/2), \tag{31}$$

we obtain from inequalities (28) and (30)

$$\alpha_\rho(\theta^\star) \in [F(\theta^\star) - 2\hat\varkappa\rho, F(\theta^\star)] \subseteq [\underline{F} - 2\hat\varkappa\rho, U] = \mathcal{A},$$

so $(\theta^\star, \alpha_\rho(\theta^\star))$ is indeed a feasible point. The distance to the initial point is bounded as follows:

$$\|(\theta_1, \alpha_1) - (\theta^\star, \alpha_\rho(\theta^\star))\|^2 \leq D_\star^2 + (U - \underline{F} + 2\rho\hat\varkappa)^2 = R^2.$$

Taking $\eta := \frac{R}{M_\rho \sqrt{N}}$ yields

$$\mathbb{E}[G_\rho(\bar\theta_N, \bar\alpha_N) - G_\rho(\theta^\star, \alpha_\rho(\theta^\star))] \leq \frac{R M_\rho}{\sqrt{N}}.$$

Lastly, the inequality (31) and Proposition 2.4 (iii) give

$$F(\bar\theta_N) - 2\hat\varkappa\rho \leq \min_{\alpha \in \mathbb{R}} G_\rho(\bar\theta_N, \alpha) \leq G_\rho(\bar\theta_N, \bar\alpha_N),$$

$$F^\star \geq \min_{\alpha \in \mathbb{R}} G_\rho(\theta^\star, \alpha) = G_\rho(\theta^\star, \alpha_\rho(\theta^\star)),$$

$$\mathbb{E}[F(\bar\theta_N) - F^\star] \leq \mathbb{E}[G_\rho(\bar\theta_N, \bar\alpha_N) - G_\rho(\theta^\star, \alpha_\rho(\theta^\star))] + 2\hat\varkappa\rho \leq \frac{R M_\rho}{\sqrt{N}} + 2\hat\varkappa\rho,$$

and the statement of the theorem follows. $\square$

## B. Additional Materials on Entropic OT

### B.1. Related Works on eOT

This subsection provides an overview of selected works on continuous entropy-regularized optimal transport. Genevay et al. (2016) tackled this problem by introducing an RKHS and optimizing the dual function (16) with SGD. This approach was extended by Seguy et al. (2018), who parameterized the dual potentials with neural networks instead of an RKHS to improve scalability. Subsequently, Daniels et al. (2021) leverage this approach to approximate the optimal transport plan, using it to develop a score-based generative model. Although this direction mostly results in computationally efficient methods that work with a general cost function, a key drawback is that small values of the regularization coefficient $\varepsilon$ cause numerical instabilities due to the exponential term in the dual objective; see Remark 3.1. The work by (Korotin et al., 2023) studies a more general formulation known as *weak OT*. The authors formulate it as a maximin problem and develop a neural-network-based algorithm under the assumption of a quadratic cost, a restriction that is later relaxed in (Asadulaev et al., 2024). However, these methods are computationally intensive due to their adversarial training nature. Mokrov et al. (2024) approach eOT from the perspective of energy-based models. Unfortunately, the resulting solver is computationally expensive as it involves iterative Langevin dynamics. Another popular approach to eOT in recent years is via the Schrödinger bridge (SB), e.g., (Gushchin et al., 2023). While SB-based solvers are also often computationally intensive, a more cost-efficient solution has been proposed by (Korotin et al., 2024). However, it relies on the quadratic cost assumption and does not support general cost. We would also like to note that a promising direction for future work is leveraging our approach for minimizing the objective (8) in (Korotin et al., 2024) to further improve scalability.

### B.2. Experiment with RKHS Representation of Dual Potentials

As mentioned earlier, LSOT (Seguy et al., 2018) is inspired by the continuous eOT approach of Genevay et al. (2016). This work considers a reproducing kernel Hilbert space (RKHS) $\mathcal{H}$ defined on $\mathcal{X}$, with a kernel $\kappa$, and applies SGD to solve the dual problem. This approach suffers from the same numerical instability as LSOT; see Remark 3.1. As an alternative, we again consider the approximation (19) of the semi-dual objective which can also be maximized by SGD. Although the variable $\alpha$ is, in general, a function of $x$, we empirically found that tuning a common scalar value $\alpha \in \mathbb{R}$ for all samples works well in the experiments described below.

Analytic form of SGD iterates for both objectives can be derived as follows. By the property of RKHS, if $u \in \mathcal{H}$, then $u(x) = \langle u, \kappa(\cdot, x) \rangle_{\mathcal{H}}$. Therefore, the derivatives of $f_\varepsilon$ take the form

$$\nabla_u f_\varepsilon(x, y, u, v) = \kappa(\cdot, x) - \exp\left(\frac{u(x) + v(y) - c(x, y)}{\varepsilon}\right) \kappa(\cdot, x),$$

$$\nabla_v f_\varepsilon(x, y, u, v) = \kappa(\cdot, y) - \exp\left(\frac{u(x) + v(y) - c(x, y)}{\varepsilon}\right) \kappa(\cdot, y).$$

Consequently, SGD iterates for the dual objective (16) can be conveniently written as

$$(u_k, v_k) = (u_0, v_0) + \sum_{i=1}^{k} \beta_i \left(\kappa(\cdot, x_i), \kappa(\cdot, y_i)\right) \tag{32}$$

$$\text{with } \beta_i := \frac{C}{\sqrt{i}} \left(1 - e^{\frac{u_{i-1}(x_i) + v_{i-1}(y_i) - c(x_i, y_i)}{\varepsilon}}\right), \tag{33}$$

where $(x_i, y_i)$ are i.i.d. samples from $\mu \otimes \nu$, and $C > 0$ is the initial stepsize. Similarly, SGD iterates for (19) are computed as follows:

$$v_k = v_0 + \sum_{i=1}^{k} \tilde{\beta}_i \kappa(\cdot, y_i),$$

$$\alpha_k = \alpha_0 - \sum_{i=1}^{k} \tilde{\beta}_i \quad \text{with } \beta_i := \frac{C}{\sqrt{i}} \left(1 - \sigma_\rho\left(\frac{u_{i-1}(x_i) + v_{i-1}(y_i) - c(x_i, y_i)}{\varepsilon}\right)\right),$$

where $\sigma_\rho(t) := \frac{e^t}{1 + \rho e^t}$.

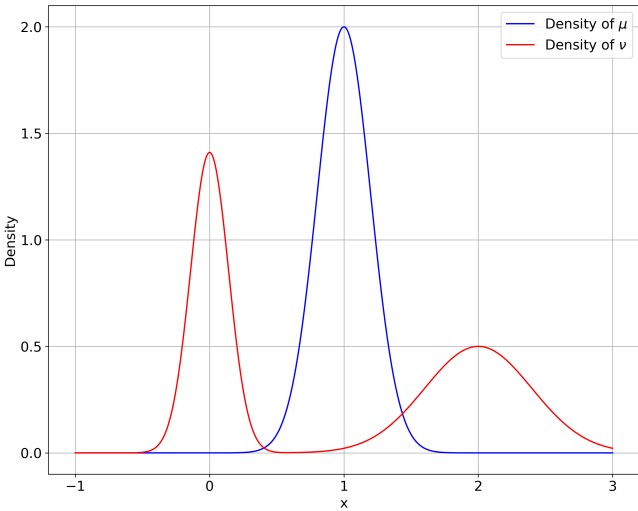

*Figure 4.* Densities of source and target distributions in the eOT experiment.

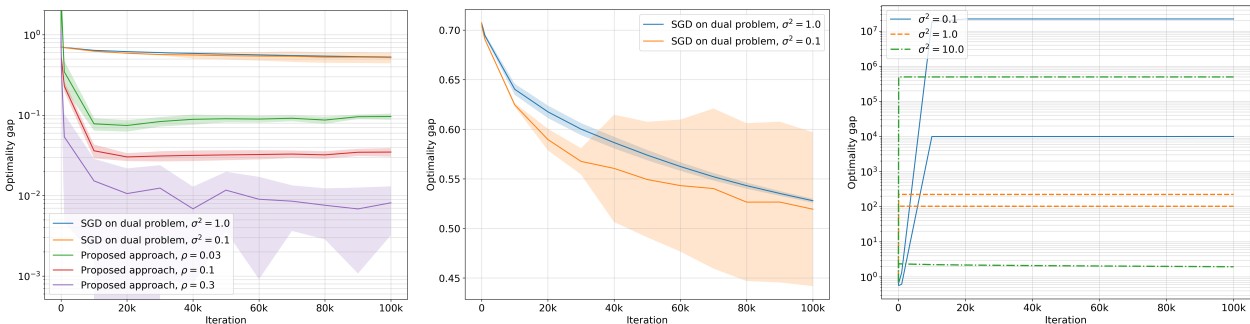

*Figure 5.* Left: convergence of kernel SGD applied to the dual objective (16) (blue and orange) and approximate semi-dual problem (18) (green, red and purple). Solid lines show average optimality gap across 20 runs, shaded regions indicate ± one standard deviation. Y-axis uses logarithmic scale. Middle: a zoomed-in view of blue and orange curves from the plot on the left. Right: examples of divergent optimality gap curves obtained by running the baseline approach with the stepsize parameter $C = 10^{-2}$.

**Experiments.** Consider a setup analogous to the one described in Section 5 of Genevay et al. (2016). Specifically, $\mu$ is a 1D Gaussian, and $\nu$ is a mixture of two Gaussians (see Figure 4 for a plot of densities). Gaussian kernel $\kappa(x, x') = \exp\left(-\frac{\|x-x'\|^2}{\sigma^2}\right)$ with a bandwidth hyperparameter $\sigma^2 > 0$ is used. The regularization coefficient is set to $\varepsilon = 0.01$. We consider kernel SGD (32) applied to the dual objective as a *baseline* approach (Genevay et al., 2016). We compare it to the proposed approach, namely, kernel SGD applied to the approximate semi-dual problem (18). For details on how the optimality gap is estimated, see Appendix B.

When applying kernel SGD to the dual and approximate semi-dual formulations, we consider hyperparameters $\sigma^2 \in \{0.1, 1, 10\}$ (kernel bandwidth), $C \in \{10^{-4}, 10^{-3}, \dots, 10\}$ (stepsize parameter), and $\rho \in \{0.03, 0.1, 0.3\}$ (approximation accuracy). Double floating-point precision is used. In the experiment, the proposed approach works best with $\sigma^2 = 10$, and $C = 1$ for $\rho \in \{0.03, 0.1\}$, $C = 10$ for $\rho = 0.3$. Baseline works best with $\sigma^2 \in \{0.1, 1\}$ and $C = 10^{-3}$. Figure 5 (left) shows performance of the two approaches. For clarity, we provide a zoomed-in view of the curves generated by the baseline in the middle. As seen from the figures, the baseline is extremely slow, which happens due to the small stepsize. Larger values of $C$ lead to numerical instabilities as illustrated by the plot on the right. This is because the exponential term can cause a large gradient magnitude at some iterations, which brings an iterate to a region where it stagnates. On the contrary, our approximate semi-dual formulation permits larger stepsizes, which results in faster convergence. Indeed, the method usually achieves a relatively low optimality gap in about $2 \cdot 10^4$ iterations, and plateaus after that.

### B.3. Computing a Proxy for Optimality Gap

Optimality gap in the experiment is estimated as follows:

*Table 3.* **KL-DRO on ACS PUMS.** Objective value (21), regression metrics, worst-group metrics, and time per epoch for the baseline (22) and the proposed estimator (24). Best results per column are shown in bold.

| Approach | Objective | RMSE ($\times 10^3$) | RMSE on hardest group ($\times 10^3$) | MAE ($\times 10^3$) | MAE on hardest group ($\times 10^3$) | Time per epoch (s) |
|---|---|---|---|---|---|---|
| Baseline | $109.3 \pm .4$ | $61.7 \pm .9$ | $75.8 \pm .3$ | $\mathbf{35.0 \pm 1.0}$ | $41.1 \pm 1.3$ | $\mathbf{92 \pm 3}$ |
| $\rho = 10^{-1}$ | $108.3 \pm .2$ | $\mathbf{60.3 \pm .8}$ | $\mathbf{74.7 \pm .3}$ | $34.9 \pm 1.0$ | $\mathbf{40.1 \pm 1.0}$ | $99 \pm 5$ |
| $\rho = 10^{-3}$ | $\mathbf{106.0 \pm .9}$ | $62.1 \pm 1.2$ | $74.9 \pm .7$ | $36.0 \pm 1.7$ | $40.9 \pm 1.7$ | $99 \pm 5$ |

1. Test sets $\{x_i\}_{i=1}^N$ and $\{y_i\}_{i=1}^N$ of size $N = 10^4$ are sampled from $\mu$ and $\nu$. The corresponding empirical distributions are denoted $\hat{\mu}$ and $\hat{\nu}$, respectively.

2. Similarly to (Genevay et al., 2016), we obtain a proxy $\hat{W}$ for $W(\mu, \nu)$ by solving the semi-discrete eOT problem

$$\max_{\mathbf{v} \in \mathbb{R}^N} \mathbb{E}_{X \sim \mu} \, \hat{h}_\varepsilon(X, \mathbf{v})$$

$$\text{with } \hat{h}_\varepsilon(x, \mathbf{v}) := \frac{1}{N} \sum_{i=1}^N v_i - \varepsilon \log\Big( \frac{1}{N} \sum_{i=1}^N e^{\frac{v_i - c(x, y_i)}{\varepsilon}} \Big) - \varepsilon,$$

which corresponds to replacing the expectation $\mathbb{E}_{Y \sim \nu}$ in (17) with the average over the test set $\mathbb{E}_{Y \sim \hat{\nu}}$. We perform 10 runs of SGD, each consisting of $2 \cdot 10^5$ iterations, and define $\hat{W}$ as the largest achieved value on the test set, i.e., the largest $\mathbb{E}_{X \sim \hat{\mu}} \, \hat{h}_\varepsilon(X, \mathbf{v})$.

3. Finally, given a potential $v \in \mathcal{C}(\mathcal{X})$, we estimate the optimality gap as $\hat{W} - \mathbb{E}_{X \sim \hat{\mu}} \hat{h}_\varepsilon(X, \mathbf{v})$, where $\mathbf{v} = (v(y_1), \ldots, v(y_N))^\top$ is the evaluation of $v$ on the test set.

# C. Additional Experimental Results for DRO

## C.1. KL-DRO under Train–Test Distribution Shift

We additionally evaluate KL-DRO on income prediction using the 2018 ACS PUMS data obtained via Folktables (Ding et al., 2021). To highlight the importance of distributionally robust training, we consider a high-income state (CA) and a low-income state (MS): 95% of CA data is allocated to the train set and 5% to the test set, whereas the corresponding ratios for MS are 10% and 90%. Thus, the train set is dominated by the high-income state, while the test set is dominated by the low-income state. Table 3 reports the objective value (21) and regression metrics for the baseline estimator (22) and the proposed estimator (24) with different values of $\rho$. Hyperparameters are set to $\lambda = 5$ and $|D| = 10$. RMSE and MAE on the hardest group denote the largest RMSE and MAE, respectively, across groups defined by the RAC1P variable (Race).

## C.2. Duality-Gap Evaluation for KL-DRO in Linear Regression

We also include a small experiment motivated by the variational form of the LogSumExp objective. For the KL-DRO objective (21), a primal iterate $\theta^k$ naturally defines the dual weights

$$p_i^k := \frac{\exp(\ell_i(\theta^k)/\lambda)}{\sum_j \exp(\ell_j(\theta^k)/\lambda)}.$$

Using the entropy-regularized variational representation of LogSumExp, the corresponding duality gap can be written as

$$\text{gap}^k = \sum_i p_i^k \ell_i(\theta^k) - \min_\theta \sum_i p_i^k \ell_i(\theta).$$

In general, the minimization over $\theta$ makes this quantity difficult to compute exactly. Therefore, we evaluate it in a simple linear regression setting with squared losses, where the inner minimization reduces to a weighted least-squares problem and can be solved in closed form.

Figure 6 reports the resulting duality gap for the proposed approach on a synthetic dataset with $n = 1000$ samples and dimension $d = 50$. The plot confirms that the proposed method steadily decreases the computable duality gap in this setting, complementing the objective-value comparisons reported in the main experiments.

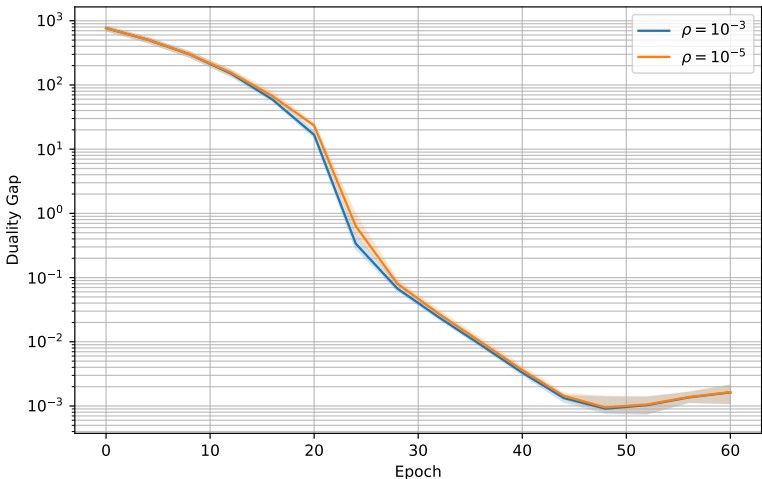

*Figure 6.* **Duality gap for KL-DRO in linear regression.** Convergence of the proposed approach for the KL-DRO objective (21) with $\lambda = 1$ on a synthetic linear regression problem with squared losses, $n = 1000$, and $d = 50$. The duality gap is computed exactly using the weighted least-squares solution associated with the softmax dual weights at each iterate. Curves show the mean over 5 seeds, and the shaded region indicates the range between the minimum and maximum values.

*Table 4.* **UOT-DRO on MNIST.** Objective value (25) at epoch 20 under 85% label noise for the baseline (26) and the proposed approximation (27). Results are averaged over 5 runs; best results per column are shown in bold.

| Approach | $\gamma = 1/5$ | | | $\gamma = 1$ | | | $\gamma = 5$ | | |
| | $\lambda = 1/5$ | $\lambda = 1$ | $\lambda = 5$ | $\lambda = 1/5$ | $\lambda = 1$ | $\lambda = 5$ | $\lambda = 1/5$ | $\lambda = 1$ | $\lambda = 5$ |
|---|---|---|---|---|---|---|---|---|---|
| Baseline | 1.92±.01 | **1.92±.01** | 1.91±.01 | 1.30±.01 | 1.26±.01 | 1.25±.01 | 0.86±.05 | 0.76±.07 | 0.59±.05 |
| $\rho = 1$ | 5.74±.59 | 6.39±.57 | 7.02±1.12 | 1.24±.34 | 0.92±.10 | 1.11±.22 | **0.67±.04** | **0.49±.02** | 0.43±.09 |
| $\rho = 10^{-1}$ | **1.43±.15** | **1.46±.48** | 1.45±.56 | 0.90±.02 | **0.83±.03** | **0.80±.04** | 0.69±.08 | 0.54±.11 | **0.43±.05** |
| $\rho = 10^{-2}$ | 1.59±.06 | 2.17±.08 | **1.53±.05** | **0.89±.03** | 0.99±.28 | 1.10±.02 | 0.68±.08 | 0.57±.10 | 0.70±.05 |

## C.3. UOT-DRO on MNIST with Severe Label Noise

We also report an additional UOT-DRO experiment on MNIST with noisy train labels. In contrast to the main experiment, here we use a very high noise ratio of **85%**. This setting is intended to test whether better objective values translate into better prediction metrics under severe label corruption.

Table 4 reports the objective value (25) at epoch 20 for the baseline (26) of (Wang et al., 2024) and the proposed approximation (27) with different values of $\rho$. Results are averaged over 5 runs, and the best result in each column is highlighted in bold.

To complement the objective values in Table 4, we report the best classification metrics achieved on the test set in Table 5. For each metric, we select the best value over all considered pairs $(\gamma, \lambda)$ separately for the baseline and for the proposed approach. We also report hardest-class metrics, defined as the worst value across classes for a fixed pair $(\gamma, \lambda)$, followed by selecting the best such value across pairs. These results show that, under severe label corruption, the improved DRO objective values are reflected in better cross-entropy, worst-class performance, and ROC AUC, although the baseline achieves higher overall accuracy.

## D. Properties of SoftPlus

Let $F(x) = \log(1 + e^{f(x)})$. Then

$$\nabla F(x) = \sigma(f(x))\nabla f(x), \tag{34}$$

$$\nabla^2 F(x) = \sigma(f(x))\nabla^2 f(x) + \sigma(f(x))\big(1 - \sigma(f(x))\big)\nabla f(x)\nabla f(x)^\top. \tag{35}$$

Suppose $f(x)$ is $L$-smooth (possibly non-convex). Let us derive smoothness constant of $F$. We will use the following

*Table 5.* **UOT-DRO on MNIST: classification metrics.** Best test-set classification metrics under 85% label noise for the baseline (26) and the proposed approximation (27). For each method, the best value is selected over all considered pairs $(\gamma, \lambda)$. Hardest-class metrics are computed as the worst value across classes for each pair $(\gamma, \lambda)$, followed by selecting the best such value across pairs. Best results per row are shown in bold.

| Metric | Baseline | Proposed approach |
|---|---|---|
| Cross-entropy (CE) | $1.36 \pm .03$ | $\mathbf{1.29 \pm .08}$ |
| CE on hardest class | $2.15 \pm .03$ | $\mathbf{1.87 \pm .07}$ |
| Accuracy | $\mathbf{0.41 \pm .04}$ | $0.34 \pm .02$ |
| Accuracy on hardest class | $0.03 \pm .01$ | $\mathbf{0.07 \pm .03}$ |
| ROC AUC | $0.88 \pm .00$ | $\mathbf{0.90 \pm .00}$ |
| ROC AUC on hardest class | $0.27 \pm .06$ | $\mathbf{0.45 \pm .06}$ |
| Avg. time per epoch (s) | $\mathbf{45.7 \pm .6}$ | $49.2 \pm 2.8$ |

**Lemma D.1.** *Consider function* $f_a(x) = \sigma(x) + 2\sigma'(x)(x - a)$, $x \geq a$ *with parameter* $a \leq 0$. *It holds* $f_a(x) \leq 2 - \frac{a}{2}$.

*Proof.* By the properties of the sigmoid function $\sigma(x)$, $\sigma'(x) \leq \frac{1}{4}$ and $\sigma(x) \leq 1$. Therefore, $f_a(x) \leq 1 + \frac{x-a}{2}$. If $x \leq 2$, the result follows. Let us now show that the derivative

$$\frac{d}{dx} f_a(x) = \sigma'(x)[3 + 2(1 - 2\sigma(x))(x - a)]$$

is negative if $x > 2$. Indeed, due to monotonicity of the sigmoid function $\sigma(x)$,

$$\sigma(x) > \sigma(2) > 0.88 \Rightarrow 2(1 - 2\sigma(x)) < -\frac{3}{2}.$$

Moreover, $x - a > 2$, so $3 + 2(1 - 2\sigma(x))(x - a) < 0$ and $\frac{d}{dx} f_a(x) < 0$. Therefore, if $x > 2$, then $f_a(x) < f_a(2) \leq 2 - \frac{a}{2}$. □

**Proposition D.2.** *Let* $f \in C^1(\mathbb{R}^d)$ *be L-smooth and bounded from below by* $f_* \in \mathbb{R}$, *then* $F(x) = \log(1 + e^{f(x)})$ *is smooth with parameter*

$$\begin{cases} \frac{4}{3}L & \text{if } f_* \geq 0, \\ \left(\frac{4}{3} - \frac{f_*}{2}\right)L & \text{if } f_* < 0. \end{cases} \tag{36}$$

*Proof.* W.l.o.g., we can assume that $f \in C^2$. From (35) and Lemma D.1 we get

$$\begin{aligned} \|\nabla^2 F(x)\| &\leq \sigma(f(x))\|\nabla^2 f(x)\| + \sigma'(f(x))\|\nabla f(x)\|^2 \\ &\leq L\sigma(f(x)) + 2L\sigma'(f(x))(f(x) - f_*) \\ &= L\big(\sigma(f(x)) + 2\sigma'(f(x))f(x)\big) - 2L\sigma'(f(x))f_*. \end{aligned}$$

Analyzing the function $h(t) := \big(\sigma(t) + 2t\sigma'(t)\big)$, one can show that $\max_t h(t) < \frac{4}{3}$. Thus, in the case $f_* \geq 0$, using the fact that $\sigma'(t) > 0$ we obtain

$$\|\nabla^2 F(x)\| \leq Lh(f(x)) \leq \frac{4}{3}L.$$

Now, consider the case $f_* < 0$. Since $\sigma'(t) = \sigma(t)(1 - \sigma(t)) \leq \frac{1}{4}$,

$$\|\nabla^2 F(x)\| \leq Lh(f(x)) - 2L\sigma'(f(x))f_* \leq \frac{4}{3}L - \frac{L}{2}f_*.$$

The claim follows.

□

*Remark* D.3. The factor $\frac{1}{2}$ in front of $-f_*$ in (36) cannot be improved. Indeed, consider $f(x) = \frac{1}{2}(x-a)^2 - \frac{1}{2}a^2$ with $f_* = -\frac{1}{2}a^2$. The second derivative of $F(x) = \log(1 + e^{f(x)})$ is

$$F''(x) = \sigma(f(x)) + \sigma(f(x))\left(1 - \sigma(f(x))\right)(x-a)^2,$$
$$F''(0) = \sigma(0) + \sigma(0)\left(1 - \sigma(0)\right)a^2 = \frac{1}{2} + \frac{a^2}{4} = \frac{1}{2} - \frac{f_*}{2}.$$

**Proposition D.4.** *If $f$ is convex, then $F(x) = \log(1 + e^{f(x)})$ is also convex.*

*Proof.* Trivially follows from (35).  □

## E. LLM Usage Disclosure

In the preparation of this manuscript, large language models (LLMs) were used to improve the readability. All substantive contributions are solely by the authors.

