# OpenReview forum: "Improved Stochastic Optimization of LogSumExp"
_ICML.cc/2026/Conference — ICML 2026 regular_

### Official Review · Reviewer_AMzP · 2026-03-01

**Soundness:** 2
**Presentation:** 3
**Significance:** 2
**Originality:** 3
**Overall Recommendation:** 2
**Confidence:** 4

**Summary:**

dealing with the stochastic optimization of the LogSumExp function is a common but tricky problem in ML. The authors propose a SoftPlus-based approximation. By tweaking the KL divergence in the dual space into a "safe KL divergence", the method keeps convexity and smoothness while letting us use SGD efficiently. It also bridges the gap with CVaR and validates things on continuous OT and DRO tasks.

**Compliance With Llm Reviewing Policy:**

Affirmed.

**Final Justification:**

These concerns have not been well discussed or addressed from the authors' responses.

**Key Questions For Authors:**

could the authors provide more experiments on high-dim, real datasets against some latest basleines, and report standard metrics like Accuracy and F1 for classification, or MSE, MAE, and R2 for regression?

could the authors clarify the hyperparameter setups like alpha,  and provide a concrete comparison of CPU time and memory usage,

**Limitations:**

a nice job but is limited in practical validation, the exp set up is unclear. baslines and datasets are really crucial issues

**Strengths And Weaknesses:**

for the positives, presentation is clear and easy to follow. from the dual perspective is a fresh take, and the connection to CVaR is a nice touch. the soundness seems to be solid with theoretical derivation of the safe KL divergence

However, there are a few weak points below.

1. the baselines are way too limited. The paper currently only compares against Eq. 19, and it could bee necessary how this stacks up against some SOTA baselines.

2. the real-world data experiments are lacking. it could be more convincing to see this tested on more real datasets, especially in high-dim scenarios, to show it actually scales in practice.

3. the evaluation metrics are kind too narrow. Simply reporting the objective function values isn't enough imo. For regression tasks, it'd be great to see MSE, MAE, or R2 scores, while for classification, Accuracy or F1 scores are pretty much required.

4. no clear statement on the convergence rate or complexity, so adding CPU time and memory usage comparisons against other algorithms is recommended

---

> ### Author Rebuttal · Authors · 2026-03-31
>
> We are grateful to the reviewer for the detailed comments and for acknowledging the novelty and theoretical soundness of our approach.
>
> Regarding the rates, one can obtain stochastic gradients for $G_\rho(\theta,\alpha)$ given in lines 203-207 (left) by differentiating the function under the integral and taking samples from $\mu$ (see, e.g., Eq.(21)). Clearly, stochastic gradients will be bounded if the loss $L$ is Lipschitz. Notably, unlike existing approaches, they will not be exponentially large. Moreover, if the loss is convex, we obtain a convex stochastic optimization problem with bounded second moment of the stochastic gradients. Thus, the standard SGD analysis leads to $O(1/\sqrt{k})$ rate, where $k$ is the iteration counter.
>
> Regarding the baselines, we in fact compare to three methods (LSOT, Eq.(19) and Eq.(23)), each coming from the particular application we consider. Furthermore, to address the questions, we conducted additional experiments to highlight potential practical utility of the proposed approach. In the UOT DRO experiment, we measured runtime and various classification metrics. Moreover, we conducted a KL DRO experiment with an MLP on a larger dataset (2018 ACS PUMS data, train size ~190k, test size ~22k, 90 features after preprocessing). A PDF with additional results can be accessed via the anonymized link: https://osf.io/t749q/overview?view_only=0488d9cf50834ae49ac8f74eae9f45e5. As for the memory complexity, the only potential overhead may come from the additional scalar variable $\alpha$. Compared to the dimensionality of the original problem, such overhead is negligible in practical scenarios.

---

> > ### Author Rebuttal · Reviewer_AMzP · 2026-04-03
> >
> > I am not completely satisfied with the response. I will keep my score.

---

### Official Review · Reviewer_XJvK · 2026-03-10

**Soundness:** 2
**Presentation:** 3
**Significance:** 2
**Originality:** 3
**Overall Recommendation:** 3
**Confidence:** 4

**Summary:**

In this paper, the authors considered the function of LogSumExp and how to optimize. They replace  the classical KL divergence by the safe KL entropy, which is an approximation of the classification of the KL divergence. In addition, they provide some theoretical results, including an equivalent variational representation of the LogSimExp function, the comparisons of the variational representation with various parameters, upper and lower bounds. Numerical  verification are provided  on entropy-regularized OT and DRO models.

**Compliance With Llm Reviewing Policy:**

Affirmed.

**Key Questions For Authors:**

- The computation remains unaffected when the dataset is large. In your model (Lines 204–207 of Subsection 2.2), updating the parameters $\theta$ and $\alpha$ requires iterating over the entire dataset $\mathcal{X}$ (gradients). Moreover, updating $\alpha$ introduces unnecessary computation, as only $\theta$ requires optimization in the LogSumExp formulation.
- The method proposed in this paper is an approximation method of the optimization of LogSumExp function.
- Equation (2): add the definition of $\mathcal{P}(\mathcal X)$.
- Lines 204--207 Subsection 2.2: change $\log\rho\to\rho$?

**Limitations:**

This paper does not solve these challenges stated in Introduction section.

**Strengths And Weaknesses:**

**Strengths:**
- This paper introduce a new approximation method for the optimization updates of LogSumExp function.
- This paper gives a clear explanation of the surrogate.
- The paper provides theoretical explanations.
- The authors evaluate the proposed methods.

**Weakness:**
- There are some mistakes which must be corrected.
- The introduction of surrogate (Safe KL entropy) solve neither challenge.

---

> ### Author Rebuttal · Authors · 2026-03-31
>
> We are grateful to the reviewer for comments, as well as for acknowledging the novelty of our approximation, its theoretical properties, and empirical evaluation. At the same time, we would be grateful if the reviewer could give more details on the listed weaknesses, especially mistakes. We would like to underline, in addition to what we have written in lines 20-38, right column, that our approach allows one to tackle both challenges listed in the introduction. First, the decision variable $\varphi$ or $\theta$ often has a large or infinite dimension. Second, the support of the measure $\mu$ can also be large or infinite. We tackle both of these challenges by an SGD-friendly approximation: there is no need to differentiate the whole sum/integral – one can just sample a batch of stochastic gradients and simply apply SGD to our formulation, since it is known that SGD has a cheap iteration. Further, stochastic gradients of logsumexp are biased and have large variance due to exponentiation. On the contrary, stochastic gradients in our approach have bounded variance. Moreover, in the experiments, we demonstrate the robustness of our approach, specifically, it works in situations where baselines lead to overflow. We can also refer to lines 20-38, right column.
>
> Unfortunately, we could not understand the reviewer’s key questions since they are formulated as statements rather than questions. Below, we comment on them based on our best understanding.
> 1. There is no need to iterate over the entire dataset X since one can apply SGD by taking only a mini-batch of stochastic gradients in each iteration (see Section 3.2, Eq.(21)). Moreover, in the finite-sum setting, our approach naturally allows using variance reduction techniques.  Importantly, $\alpha$ is a univariate parameter, and hence its update does not influence the computations much compared to the update of $\theta$ – a reasonable price for the possibility to use SGD-type methods without bias and with bounded variance.
> 1. Yes, we propose a universal approximation to the logsumexp function and our approximation allows us to use SGD-type methods with more stability and faster convergence.
> 1. $\mathcal{P}(X)$ is defined in the “Notation” subsection in lines 69-70, right column.
> 1. This equation is correct since, as we write above it, we shifted $\alpha$ by $\log \rho$ compared to (6).

---

> > ### Author Rebuttal · Reviewer_XJvK · 2026-04-03
> >
> > I thank the authors' reply. My concern do not solve at all.

---

### Official Review · Reviewer_2fZk · 2026-03-12

**Soundness:** 3
**Presentation:** 3
**Significance:** 3
**Originality:** 3
**Overall Recommendation:** 4
**Confidence:** 4

**Summary:**

The paper proposes an approximation for stochastic optimization of the LogSumExp (LSE) loss function that tries to address the problem of computing gradients of a large sum appearing in LSE. The approximation preserves the convexity and smoothness properties & can be used as a plug-in replacement for a number of applications in ML using LSE. While similar approximations of LSE exist, the particular safe-KL approximation enabling unbiased stochastic gradients appears novel.

**Compliance With Llm Reviewing Policy:**

Affirmed.

**Final Justification:**

The paper proposes a stochastic approximation to the log-sum-exp (LSE) operator with potential applications across several ML settings. I find the idea novel and the theoretical development largely sound.

In my initial review, I raised concerns regarding presentation, empirical validation, and theoretical clarity in certain applications (e.g., unbalanced OT and Safe-KL). The authors have responded constructively by providing additional experiments, including runtime measurements and results on a larger-scale dataset. These additions are helpful and partially address my concerns.

The approach could be impactful if it consistently delivers computational savings without degrading downstream performance. The computational advantage is demonstrated only in a limited setting, and more extensive comparisons across downstream tasks would be important to substantiate the practical benefits of the method.

Overall, the rebuttal improves the paper but does not fully resolve my concerns about empirical validation in downstream settings. I believe the work is promising, however, in the current form, I would stay with 'weak accept'.

**Key Questions For Authors:**

1. To justify the main claims of the paper, (i) there should be a computation time comparison of the proposed approach compared to other approximations of LSE (ii) empirical performance comparison on downstream applications.
2. Can there also be a duality gap comparison to check the efficacy of the proposed optimizer?

**Limitations:**

There is no separate discussion on limitations, the paper only mentions about the future work of expanding the empirical section.

**Strengths And Weaknesses:**

Soundness: The main results & theoretical proofs appear essentially correct.
Presentation: The presentation appears somewhat redundant. For eg. Figure 1 was perhaps not needed as it is clear from the mathematical expression what f_\rho(t) converges to. Similarly, Sec 3 seems to have redundant re-writing of formulations. Having proof sketches in the main paper will also help the reader.
Significance: The proposed stochastic approximation of LSE can benefit various ML applications. The significance of the approach can better be understood if the empirical downstream performance stays the same (with reduced computation time) or improves. Especially, in problems like unbalanced OT, it is not clear if safe-KL can mimic the performances with traditional KL. This is because KL-regularized UOT is theoretical motivated as a metric (under sufficient conditions on the cost & the regularization hyperparameters). Simply replacing KL with Safe-KL need not satisfy the metric properties. It might also be interesting to see if the approach can provide a sound approximation of Info-NCE loss for contrastive learning.
Originality: While the proof techniques are not new, the proposed approach appears novel.

---

> ### Author Rebuttal · Authors · 2026-03-31
>
> We would like to thank the reviewer for constructive suggestions for improving the presentation and for acknowledging the potential and novelty of our approach, as well as the soundness of our main results and theoretical proofs.
>
> Indeed, more theoretical justification is needed for proposing the Safe version of UOT. This is an important direction for future work, whereas the current paper is a good starting point in this direction.
> Currently, the approximation error bound we derived directly implies guarantees on the optimality gap in the applications we consider, and the experimental results empirically validate the approximation properties.
> At the same time, we agree that a detailed study of other theoretical aspects of Safe KL-based UOT (and other applications) is an important direction for future work. We also agree that additional directions like approximating Info-NCE would be very interesting.
>
> To address the questions, we conducted additional experiments. In the UOT DRO experiment, we measured runtime and various classification metrics. Moreover, we conducted a KL DRO experiment with an MLP on a larger dataset (2018 ACS PUMS data, train size ~190k, test size ~22k, 90 features after preprocessing). A PDF with additional results can be accessed via the anonymized link: https://osf.io/t749q/overview?view_only=0488d9cf50834ae49ac8f74eae9f45e5.
>
> In the majority of practical scenarious, the duality gap isn't readily computed, so the convergence is assessed in comparison with a baseline. At the same time, the derived bound on the approximation error and the standard SGD convergence analysis yield theoretical guarantees for the optimality gap.

---

> > ### Author Rebuttal · Reviewer_2fZk · 2026-04-02
> >
> > Thank you for the response and the additional experiments. I would like to stay with my score.
> > I think the computational time comparison is shown in only one of experiments. More extensive evaluation in terms of further downstream application scores would benefit the work. I also think specifically for LSE objectives, there is an explicit variational structure via entropy-regularized formulation that could certify duality gap in terms of KL divergence.

---

> > > ### Author Response · Authors · 2026-04-08
> > >
> > > We thank the reviewer for the additional remarks. In fact, the computational time comparison was shown for two experiments (KL DRO and UOT DRO). As for the eOT experiment, one run of the baseline takes 301 seconds (std 2s), while one run of the proposed approach takes 282s (std 2s). Regarding the duality gap, suppose a primal iterate \(\theta^k\) is given, and the dual iterate is defined by $p^k_i := \frac{\exp(\ell_i(\theta^k)/\lambda)}{\sum_j \exp(\ell_j(\theta^k)/\lambda)}$. Then, the duality gap can be shown to have the form $\Delta^k=\sum_i p^k_i \ell_i(\theta^k)- \min_{\theta} \sum_i p^k_i\,\ell_i(\theta)$. Due to the minimization w.r.t. $\theta$, the duality gap isn't readily available in most practical scenarios, including our experiments, where neural networks are used. Still, we plot the duality gap for our approach in a simple linear regression experiment, where it can be computed exactly. The plot can be found under the same link, https://osf.io/t749q/overview?view_only=0488d9cf50834ae49ac8f74eae9f45e5.

---

### Official Review · Reviewer_7WCM · 2026-03-12

**Soundness:** 3
**Presentation:** 3
**Significance:** 2
**Originality:** 3
**Overall Recommendation:** 4
**Confidence:** 4

**Summary:**

This paper studies a new approximation to the KL divergence, which they call the safe KL divergence. The dual of the new safe KL divergence has a form that contains a rescaled SoftPlus rather than the LogSumExp. This can be viewed as an approximation to LogSumExp that yields more efficient optimization properties. Experiments demonstrate superior stability for small regularization entropic optimal transport, the ability to handle smaller batches in DRO with the KL divergence, and better convergence for DRO with unbalanced optimal transport.

**Compliance With Llm Reviewing Policy:**

Affirmed.

**Final Justification:**

I maintain my original rating. The authors addressed most of my concerns; however, I still don't see a coherent narrative connecting their theory to their experiments. Therefore, I will maintain my score of weak accept.

**Key Questions For Authors:**

1. I am unsure how the experiments connect to the theory, and this needs to be made clearer. Are you arguing for stability, convergence rate, or something else?
2. What driving applications are envisioned for this method? It seems that entropy regularized OT is a key driver, so in that case, can you derive more theoretical properties that describe the advantage of this approximation?
3. I am a bit lost on how exactly this is solving the problem of an infinite number of terms in the dual, as is mentioned in the abstract. Can the authors make this crystal clear? Doesn't the dual still involve an integral of a log of a function of x?

**Limitations:**

The authors have not discussed any limitations or disadvantages of their methodology - they should spend some time doing this, maybe in a short subsection.

**Strengths And Weaknesses:**

Strengths:
- The paper is technically sound, offering a straightforward analysis of a new approximation to the LogSumExp.
- The paper is clearly written and easy to follow.
- As the authors mention, there is no universal way to treat the LogSumExp functional, and so from that perspective, this represents a novel formulation for general optimization improvement.
- The experiments demonstrate that this may offer key algorithmic advantages for KL regularized problems.

Weaknesses:
- The significance of the paper is a bit lost. The abstract seems to sell this as an approximation that makes stochastic optimization more efficient, but the experiments seem to demonstrate superior stability. A clearer discussion of the significance of the proposed method along various axes (speed, stability, accuracy, etc) is needed.
- Counter to the originality, it does seem that a similar proposal for modifying KL was considered previously in the context of CVaR.

---

> ### Author Rebuttal · Authors · 2026-03-31
>
> We are grateful for the thoughtful review that helps us improve the paper, as well as for acknowledging soundness, clarity and novelty of our work.
>
> Answers to key questions.
> 1. Our approach allows for improvement in several axes mentioned by the reviewer. The first is indeed stability. Our approximation allows one to avoid overflows when applying optimization algorithms, which is demonstrated in our experiments where baselines stop working at some point due to floating point exceptions. The second is the speed, since stability means that we can use larger stepsizes leading to faster convergence. Moreover, our controlled bias enables smaller batches, which means faster convergence in terms of samples.
> 1. We consider our approach to be universal for logsumexp/log partition minimization that can be found in many applications, such as extreme classification, partial AUC maximization, Distributionally Robust Optimization, and others. Among them, one of the important examples is the OT, where this minimization appears due to entropic regularization. We can easily derive the stochastic gradient in (15) by differentiating (16) and sampling from $\mu,\nu$, and estimate the second moment of this stochastic gradient. Notably, it will depend only on the derivatives of the parametrizations of $v,\alpha$, and, unlike existing approaches, will not have exponential growth. By applying standard SGD analysis, we obtain the convergence rate of the order $1/\sqrt{k}$ after $k$ iterations. Using the analysis from (Daniels et al. *Score-based Generative Neural Networks for Large-Scale Optimal Transport*), we can also derive an explicit expression for the transportation plan.
> 1. Yes, our approximation will still involve an integral of a log of a function of x, see (6). Yet, since the integral is outside of the logarithm, this becomes the standard stochastic optimization problem, where an expectation of a logistic-loss-like objective is minimized. By sampling from $\mu$, we obtain unbiased stochastic gradients with a bounded variance, unlike the original log-partition problem formulation.

---

> > ### Author Rebuttal · Reviewer_7WCM · 2026-04-02
> >
> > The authors have mostly addressed my questions, but my main sticking point remains 1, and in particular, trying to find a coherent narrative for the paper. In particular, can the authors give theoretical statements/evidence for the observed practical gains?

---

> > > ### Author Response · Authors · 2026-04-08
> > >
> > > We thank the reviewer for the additional question. Regarding the theoretical convergence, we state the following proposition (we will include the proof in the paper):
> > >
> > > **Proposition.** Let the problem in lines 204-207 (left) satisfy the following conditions: the loss $L(x,\theta)$ is convex w.r.t. $\theta$ and has bounded gradients, $\|\nabla_\theta L(x,\theta)\|\leq G$ for all $x \in \mathcal{X},\theta\in \Theta$, and $\Theta$ is a compact convex set. Then, Stochastic Mirror Descent with the Euclidean proximal setup applied to that problem generates a sequence $(\theta_k,\alpha_k)$ such that $\mathbf{E} G(\theta_k,\alpha_k)-G(\theta^{\star},\alpha^{\star})\leq \text{diam}(\Theta) G/(\rho\sqrt{k})$.
> > >
> > > Note that due to Proposition 2.4, the same convergence rate, up to $O(\rho)$ additive error, applies to the original problem. Unlike our approach, the existing methods suffer from either a large bias, or exponential constants in the convergence rates. For example, Proposition 1, Eq.(11) in *Large-scale methods for distributionally robust optimization (2020)* by Levy et al. estimates the bias arising when LogSumExp over a batch of size $n$ is used for stochastic gadient computation. The bias scales as $\frac{1}{n}$ and may be large when moderate-sized batches are used. Another example is Theorem 3 in *Outlier-Robust Distributionally Robust Optimization via Unbalanced Optimal Transport (2024)* by Wang et al., describing convergence of SGD applied to an objective with exponentials. As seen from its proof, the convergence rate involves **exponentials** of bounds on gradients and function values, that may be prohibitively large in practice.
> > >
> > > To sum up, our approach avoids the dependence of the bias on the batch size, as well as the exponential constants in the convergence rate, thereby providing both faster and more stable convergence.

---

### Decision · Program_Chairs · 2026-04-30

**Decision:**

Accept (regular)

**Comment:**

This paper introduces a "safe KL divergence" approximation for the LogSumExp function. This framework provides unbiased stochastic gradients with bounded variance , addressing core computational bottlenecks in distributionally robust optimization and optimal transport.

Strengths: The formulation is mathematically rigorous and original. It successfully preserves convexity and smoothness for SGD , prevents floating-point overflows , and ensures stable convergence rates ($O(1/\sqrt{k})$) without problematic exponential constants.

Weaknesses: The empirical validation is narrow. Reviewers highlighted a lack of comparisons against state-of-the-art baselines and noted that real-world scalability on complex downstream tasks remains underexplored, even with the rebuttal's additions.

Conclusion: Although the experimental benchmarking is limited , the theoretical solution to this fundamental optimization challenge is mathematically sound and valuable. Hence I recommend a weak acceptance.